# Landscape to microhabitat: Uncovering the multiscale complexity of native and exotic forests on Terceira Island (Azores, Portugal)

Sébastien Lhoumeau[1]*, Rui B. Elias[1], Dominik Seidel[2], Rosalina Gabriel[1], Paulo A. V. Borges[1,3,4]

**1** cE3c- Centre for Ecology, Evolution and Environmental Changes/Azorean Biodiversity Group & CHANGE – Global Change and Sustainability Institute, School of Agricultural and Environmental Sciences, University of the Azores, Azores, Portugal, **2** Department for Spatial Structures and Digitization of Forest, Faculty of Forest Sciences and Forest Ecology, University of Göttingen, Göttingen, Germany, **3** IUCN SSC Atlantic Islands Invertebrate Specialist Group, Angra do Heroísmo, Azores, Portugal, **4** IUCN SSC Species Monitoring Specialist Group, Angra do Heroísmo, Azores, Portugal

\* seb.lhoumeau@gmail.com

## Abstract

This study aims to identify the structural and compositional differences between native and exotic woodlands on Terceira Island, Azores. Based on landscape, habitat, and microhabitat analyses, remnants of native forests appeared to be associated with less accessible terrains. A more homogeneous structural complexity is exhibited, derived from the numerous branching patterns of the endemic vascular plant species. In contrast, exotic forests exhibit structural heterogeneity driven by mixed non-indigenous vascular plant species as a result of human actions such as afforestation and latter invasion of exotic tree species, after abandonment of the agricultural use. The ground and canopy layers in exotic forests were more invaded by non-indigenous species, while the understory demonstrated greater resilience by being mostly composed of indigenous species. Our findings highlight the structural and ecological differences between native and exotic woodlands, reflecting the historical transformation of forest cover in the Azores. These insights emphasize the importance of long-term monitoring and structural assessments in informing conservation efforts aimed at preserving native forests and managing invasive species in exotic woodlands.

## Introduction

Forests play a critical role in maintaining global biodiversity [1,2], regulating the carbon cycle [3,4], and providing a wide range of ecosystem services [5–7] as a result of their structural complexity [7–10]. Forest structure is defined by several components like spatial organization of the trees and shrubs [11], vertical foliage distributions [12], horizontal canopy distribution [13] and amount of dead wood [14,15]. Tree and forest structure can serve as an indicator of tree vitality [16,17] or forest

**Data availability statement:** Data are available in supplementary materials.

**Funding:** S.L. is funded by the Azorean Government Ph.D. grant numbers M3.1.a/F/012/2022. R.B.E, R.G., and P.A.V.B. are currently funded by the projects Azores DRCT Pluriannual Funding (M1.1.A/INFRAEST CIENT/001/2022) and FCT—Fundação para a Ciência e Tecnologia FCT-UIDB/00329/2023 DOI 10.54499/UIDB/00329/2020. D.S received no outside funding.

**Competing interests:** The authors have declared that no competing interests exist.

and ecosystem dynamics, helping to track changes driven by natural processes or human activities [10,18]. The structure of forests around the globe is influenced by a multitude of environmental factors, both local and global, including climate, soil characteristics, and species composition, making them complex ecosystems to study [19].

Among the most vulnerable ecosystems are forests found on oceanic islands, where isolation and unique environmental pressures create singular forest dynamics [20,21]. Island ecosystems are often referred to as "natural laboratories" due to their isolation, which leads to the evolution of unique species, communities and ecological interactions. However, these ecosystems are highly sensitive to disturbances such as habitat degradation, invasive species, and climate change [22]. Understanding the structure of forests in island environments is therefore of high relevance both theoretically (see for example [23]) and for conservation purposes (see for example [24]), as these forests tend to house endemic species that rely on specific ecological conditions [25].

Characterising forest structures in island ecosystems like those of the Azores archipelago is therefore crucial for managing biodiversity and assessing ecosystem health. Despite the availability of some work on the environmental drivers determining the forest structure in the neighbouring archipelago of Canary Islands (e.g., [26,27]), few data is available on Azorean forests (see [4]). The Azores, one of the most remote archipelagos in the North Atlantic Ocean, is made up of nine islands shaped by both natural processes and six centuries of human influence [28–30]. These islands are known for their rich biodiversity, unique ecosystems, and varying vegetation types, which include both native and exotic woodlands [28,31].

The native vegetation, part of the archipelago's natural heritage, consist of several vegetation types [28,32–35] and are home to many endemic taxa [36], not only of vascular plants but also bryophytes [37] and arthropods [38]. However, historical descriptions and paleoecological analyses [30,39–42], provide evidence of the extent and progression of land-use changes since human settlement [43]. Large areas being replaced by exotic species such as *Cryptomeria japonica* (L. f.) D.Don and *Pittosporum undulatum* Vent., introduced for commercial forestry [29]. According to the State of the Environment Report of 2022 (Azorean Regional Government), based on a forest inventory from 2007, forest areas occupied 30% of the land surface in the Azores. From these, 31,1% are production forests; 33,4% are *Pittosporum* dominated woodlands; and 35,5% are natural and seminatural areas (including native forests). i.e., approximately ⅓ for each type. As a result, the Azores's landscape has several patches of native and exotic woodlands in a matrix of agricultural fields [32,43–45], each contributing differently to the current ecological dynamics of the island. Native vegetation are crucial for maintaining the Azorean island's biodiversity, acting as refuges for endemic plants and animals [46]. In contrast, Azorean exotic woodlands present a different spatial structure, often lacking the structural complexity and diversity of native forests [47] but see [48]. Understanding how these woodlands differ in terms of structure is essential for assessing their ecological function and resilience to environmental disturbances.

Terceira Island, the third largest of the nine islands in the archipelago is considered to have the best-preserved and most representative native vegetation patches [28,49]. As such, it provides an ideal setting for studying the defining characteristics and structural complexity of both native and exotic woodlands. This study is based on a global and holistic approach to characterise structural complexity of the woodlands. We use LiDAR (Light Detection And Ranging) technology to analyse the structural characteristics of these ecosystems. Furthermore, the use of remote sensing data from satellites and Unmanned Aerial Vehicle (UAV) allows for the acquisition of more detailed information regarding the canopy layer. Finally, the vegetation data collected from surveys provides information on species composition and density across all layers. The data, gathered on Terceira Island, are used to address the following question: What are the defining characteristics of the vegetation structure of the various types of Azorean forests, with a particular focus on the native and exotic woodlands? By answering this question, this study aims to provide a detailed characterisation of Terceira's forests, offering insights into how both native and exotic forest types are integrated within the island's broader ecosystem.

## Materials and methods

### Study area and site selection

This study was conducted on Terceira Island, part of the Azores Archipelago (Portugal, Macaronesia), located in the North Atlantic Ocean approximately 1,500 km west of mainland Portugal (Fig 1). To assess and compare the vegetation structure of native and exotic forests, a total of 18 sites of 20 x 20 m (400m²) were selected (Table S1 in S1 Data). Nine of these sites are located within areas of core native vegetation which are dominated by *Juniperus brevifolia* (Seub.) Antoine and *Ilex azorica* Gand. The remaining nine sites were situated in areas dominated by the invasive *Pittosporum undulatum*. Of the 18 sites, 13 are part of the long-term project SLAM (Long Term Ecological Study of the Impacts of Climate Change

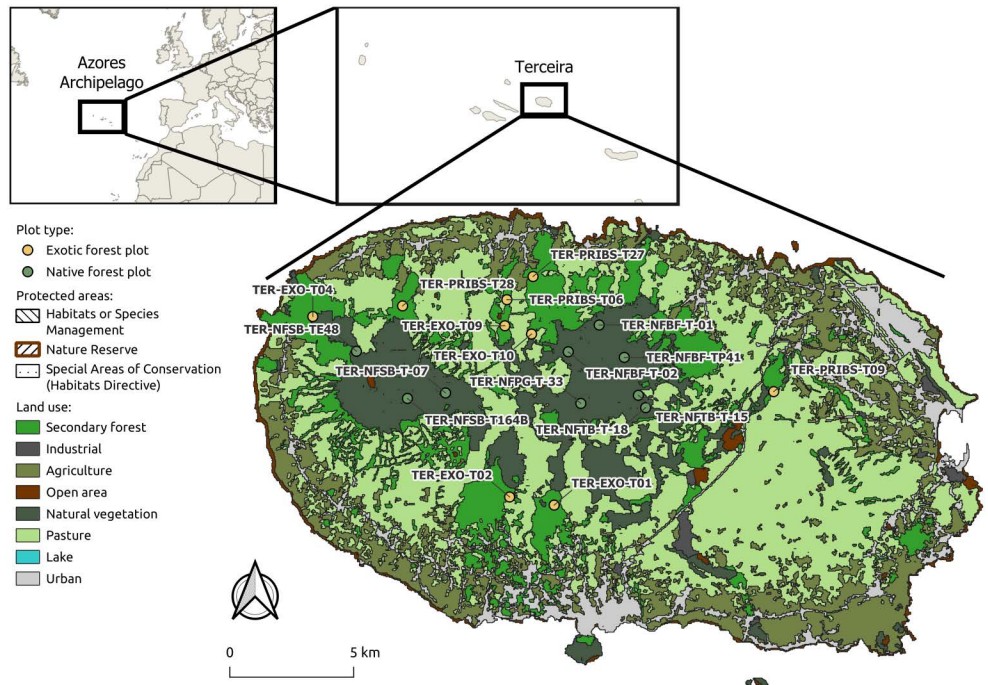

**Fig 1. Location of sampling sites.** Map of the study area on Terceira Island, Azores, showing the location of the 18 forest sites surveyed, including nine within native forest and nine within exotic woodlands. Protected areas are indicated based on data from [50], and land use classifications are provided by the Azorean Government.

in the natural forest of Azores) that started in 2012. Focused on arthropods monitoring, it aims to understand the impact of the drivers of biodiversity erosion on Azorean native forests [51–53].

## Data acquisition

All the necessary permits for data collection were obtained from DRAAC (Direção Regional do Ambiente e Ação Climática) for access to sites in protected areas and from AAN (Autoridade Aeronáutica Nacional) and SRAAC (Secretaria Regional do Ambiente e Ação Climática) for UAV data collection.

### Terrestrial laser scanning

LiDAR is a close-range remote sensing tool that produces high-resolution, three-dimensional data on canopy height, vegetation layers, and tree density, making it ideal for forest structure analysis. Here we used a Faro Focus M70 (Faro Technologie Inc., Lake Mary, USA) terrestrial laser scanner mounted on a tripod and operated in single-scan mode to characterise the vegetation structure with established 3D metrics. While the ground-based approach of terrestrial laser scanning is less applicable to large areas when compared to airborne laser scanning, it provides objective and highly-efficient spatial data with very high-resolution even in very dense and structurally complex forests [54–56].

The device is a phase-shift-based 3D laser scanner capturing a field of view of 305–360 degrees vertically and horizontally, respectively. It captures objects in a distance of up to 70 m that reflect the emitted laser light (1050 nm wavelength) at their surface and stores the object's local coordinates as well as the intensity of the reflected signal. Mounted on a standard tripod at breast height (1.3 m) the scanner rotates and captures the entire scenery in its surroundings creating a so-called point cloud. Using the software Faro Scene (Faro Technologies Inc. Lake Mary, USA), the scan data was filtered for erroneous measurements and converted into txt-file format for further calculation of several forest structural metrics. To capture the forest plots, we conducted five scans per plot. One in the centre and one in each corner of the plot (five-on-a-dice setup) as conducted in earlier studies (e.g., [19,56]. The values of the metrics were ultimately averaged to derive a single mean value for each metric in each plot.

We derived several established metrics for single-scan laser scanning, namely the Stand Structural Complexity Index (SSCI, see [57] for details) including the mean fractal dimension (abbr. mean_FD) and the effective number of layers, the understory complexity index (UCI, see [58] for details), canopy openness (see [59] for details), foliage height diversity (after [60] and adapted to laser scanning as described in [61]). Effective number of layers and vertical evenness were calculated according to [62].

### UAV canopy mapping

To assess the fine-scale structure of the forest canopy, we employed the use of an unmanned aerial vehicle (UAV), specifically a DJI Mavic Pro drone, during summer 2024 and in temporal proximity to the laser scanning. Flight missions were planned and executed using the Litchi Mission Hub software, which allowed for precise control over the UAV's trajectory. The flight plans were designed to cover an area of 30 x 30 meters following a boustrophedon pattern. We slightly expanded the mapped area to minimize distortion effects during the reconstruction process. This approach ensured high-quality RGB imaging with a targeted ground sampling detail of 1 cm per pixel, enabling detailed analysis of the canopy surface.

The collected imagery was processed using Agisoft Metashape (version 1.5.2) software to generate a dense point cloud, providing a 3D representation of the canopy structure. The roughness of the canopy surface was analysed using CloudCompare (version 2.13.2). In this analysis, the "roughness" value was calculated as the distance between each point in the point cloud and the best-fitting plane computed based on its nearest neighbours (radius = 20 cm). The roughness of a structure refers to how smooth or bumpy its surface is. A surface with high roughness has more bumps and

irregularities, while a low-roughness surface is smoother. Finally, the point cloud data was rasterized to create a spatially continuous dataset, enabling further quantitative analysis of canopy structure across the study sites.

To quantify the roughness over the plot area, we used QGIS (version 3.34.9) to calculate the average and standard deviation of the roughness values across a 400 m² area for each site. This allowed for a more comprehensive assessment of canopy surface variation within the different forest types, providing insights into structural heterogeneity across the native and exotic forests.

## Vegetation survey

This survey aimed to collect detailed data on forest species composition across the study plots. The exotic forest plots were surveyed in the spring of 2023 within 20 x 20 meter plots, while the native forest plots were surveyed earlier, during the summer of 2012 and 2013, within larger 50 x 50 meter plots. Data on woody species within native plots are taken from [63].

Prior to data collection, we conducted a brief field reconnaissance of each site. Although not formally quantified, this visual assessment confirmed that species composition and distribution were remarkably consistent across the entire area of each plot and its surroundings, supporting our use of differing plot sizes without introducing bias in structural complexity estimates. The methodology was firstly designed and implemented in 2012 during earlier projects, notably ISLANDBIO-DIV and MOVECLIM. The formal publication of the protocol included contributions from multiple specialists and research groups leading to a standardised survey protocol for island biotas [64].

Several important variables were measured during the surveys to assess the forest structure. Species richness ($H_0$), representing the total number of species present in each plot [65], was recorded to evaluate diversity within the forest plots. The basal area of trees with a diameter at breast height (DBH) greater than 10 cm was calculated per hectare. This measure is important assessing the importance of tree species in the overall forest structure. Finally, we counted the number of all woody vascular plants per species and per square metre in five subplots of 5 x 5 metres, placed in a five-on-a-dice arrangement, which allowed us to assess the vegetation.

These measurements were further classified in three distinct forest strata: the ground layer, the understory, and the canopy, offering a vertical profile of the forest structure. The classification of species as endemic, native, introduced, or invasive follows the regional checklist [36] and updated data from AZORES BIOPORTAL (https://azoresbioportal.uac.pt/). Invasive status is further based on species listed in Decreto-Lei n.º 565/99, de 21 de Dezembro, which identifies legally recognized invasive plants in Portugal, including the Azores. It results in a total of twelve variables, with each stratum and species category providing specific insights into the forest's ecological composition and structure.

## Sites and landscape characteristics

For each of the 18 study sites on Terceira Island, detailed data on elevation and landscape characteristics were collected. Elevation data for each site were extracted using digital elevation models sourced from the Shuttle Radar Topography Mission (SRTM) [66], providing a precise measurement of the elevation at which each plot is located. Moreover, terrain slope was assessed within a quadrat of side 500m using Horn's method [67] implemented in QGIS software (version 3.34). Mean, minimum, maximum and standard deviation of the slope were assessed.

Furthermore, landscape heterogeneity was quantified in the surrounding area of each site. Land use data, obtained from the Azorean government (https://ot.azores.gov.pt/), were processed and analysed in QGIS. A quadrat of side 500 meters was designed around all plots, and the proportion of land use was assessed per category.

Slopes and land use were computed considering a spatial resolution of 30 meters per pixel.

## Data analysis

The dataset used for data analysis is composed of a total of 49 variables describing several features of the forests grouped accordingly to their ecological relevance. Variables are described in the Table S1. All data management and

preprocessing steps were conducted using Julia version 1.10 [68], while the NMDS computations and correlation analyses were performed in R version 4.4.1 [69]. This workflow ensured efficient data handling and precise statistical analysis of the complex; multi-variable dataset collected across the study sites.

Patterns in forest structure and composition across the various study sites, was explored using 2D Non-metric Multidimensional Scaling (NMDS) analysis. The NMDS were calculated using the metaMDS function from the vegan package in R [70], based on Bray-Curtis dissimilarity, which is suitable for ecological data with multiple variables. This approach enabled us to simplify the data set while maintaining the underlying ecological relationships, thus facilitating a clear visualisation of the differences between native and exotic forest plots.

A global NMDS was conducted, with all the 49 variables considered. This analysis was complemented by an Analysis of Similarities (ANOSIM) test using the anosim function from the vegan R package, with the objective of evaluating the statistical significance of differences between native and exotic forest plots. ANOSIM is a non-parametric test that is used to compare the dissimilarities between predefined groups. This is achieved by ranking the pairwise distances and generating an R statistic. An R value close to 1 indicates high dissimilarity between groups, whereas an R value near 0 indicates low or no difference. For this study, we used the Bray-Curtis dissimilarity computation and 999 permutations, the two groups considered are the forest types (namely native and exotic forests). Subsequently, for each site, we analyzed the relationship between the position of the plots in the 2D NMDS space and their corresponding traits in the original 49D trait matrix. This was done by computing Spearman's rank correlation test, implemented via the Hmisc package in R [71]. By correlating the 2D coordinates of the plots in the NMDS space with the original multidimensional trait data, we identify the variables most strongly associated with variations in the forest structure.

Furthermore, each variable was assigned to a group based on its potential ecological explanatory meaning hereafter called traits by following the definition from [72].The following Fig 2 provides a visual representation of the clustering and spatial hierarchical links between the traits. For each of the traits, a NMDS was performed to investigate the ways in which each forest type differs according to the spatial scale under consideration.

## Results

A global perspective points out that the selected native and exotic plots on Terceira Island exhibited divergent species compositions. Vegetation surveys underscored the prevalence of endemic species in native plots, with most vascular plants occurring in the understory (57 species) and ground strata (49 species), exhibiting notable contributions from

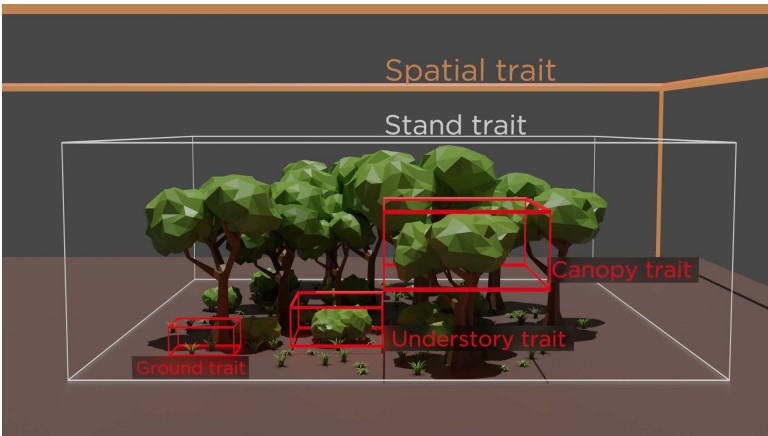

**Fig 2. Visual representation of the group of variables considered.** The organisation is based on the spatial extent from landscape scale (Spatial traits) to plot scale (stand traits) micro-habitat (canopy, understory and ground traits).

endemic and native species (Table 1). By contrast, exotic plots exhibited a more balanced distribution across the strata, with notable numbers recorded in the understory (22 species) and canopy (15 species) (Table 1). Further detailed data can be found in table S5 in S1 Data.

When added together with the other variables within the global NMDS analysis (Fig 3), we found significant structural differences between native and exotic forest plots. This discrimination of the forest type is visible on the first axis of the NMDS. The ANOSIM resulted in an R statistic of 0.9036 and a significance level of $p < 0.001$. This high R value, close to 1, indicates strong dissimilarity between the two forest types, with minimal overlap in structural characteristics.

The clustered NMDS analysis based on the forest traits (Fig 4) demonstrated that the plots could be distinctly separated into two groups, corresponding to the forest types in all cases. However, the magnitude of differentiation (R statistics of the ANOSIM) differs according to the trait considered. The greatest differentiation of forest types was obtained using stand traits, while the least differentiation was obtained using ground community traits. Interestingly, at the micro-habitat scale, a gradient of differentiation was observed following the verticality of the layers (Fig 4).

The Table 2 presents the statistically significant variables correlated with the axis of the corresponding NMDS for each trait.

The native forest plots (Fig. 5) were in more complex terrain, characterised by high elevation and slope, and in large, homogeneous areas surrounded by natural vegetation. These stands were structurally complex, principally composed of large and old endemic trees with reduced height. A detailed investigation of the micro-habitats within each layer revealed a structural and compositional homogeneity in the canopy, as well as a high level of pristine composition when considering the endemic tree species. The understory was found to be highly structurally complex, with a diverse range of indigenous (native and endemic) plant species. Finally, the ground layer is mainly occupied by indigenous vascular plants.

In contrast, exotic forest plots (Fig. 5) were in areas with more convenient access, situated within a mosaic of human-influenced landscapes. The stands exhibited a high and multi-layered structure. It is also noteworthy that, in comparison to native forest plots, these plots exhibited greater heterogeneity in their structure and composition. This heterogeneity is also evident in the canopy layer. Despite the dominance of introduced and invasive species, the canopy openness was also identified as a statistically significant variable associated with the separation of plots. The understory was dominated by native species and exhibited a lower structural complexity compared to the native forest plots. The ground layer demonstrated a clear dominance of introduced species.

**Table 1. Number of vascular plant species recorded in exotic and native plots according to their dominant strata and their colonisation status from [36].**

| Dominant strata | Colonisation status | Number of species in exotic plots | Number of species in native plots |
|---|---|---|---|
| Ground | Introduced | 4 | 1 |
| | Native | 17 | 49 |
| | Endemic | 3 | 54 |
| | Unknown | 2 | 0 |
| Understory | Invasive | 16 | 13 |
| | Native | 22 | 57 |
| | Endemic | 16 | 41 |
| | Unknown | 1 | 0 |
| Canopy | Invasive | 15 | 1 |
| | Introduced | 7 | 1 |
| | Native | 4 | 1 |
| | Endemic | 13 | 68 |

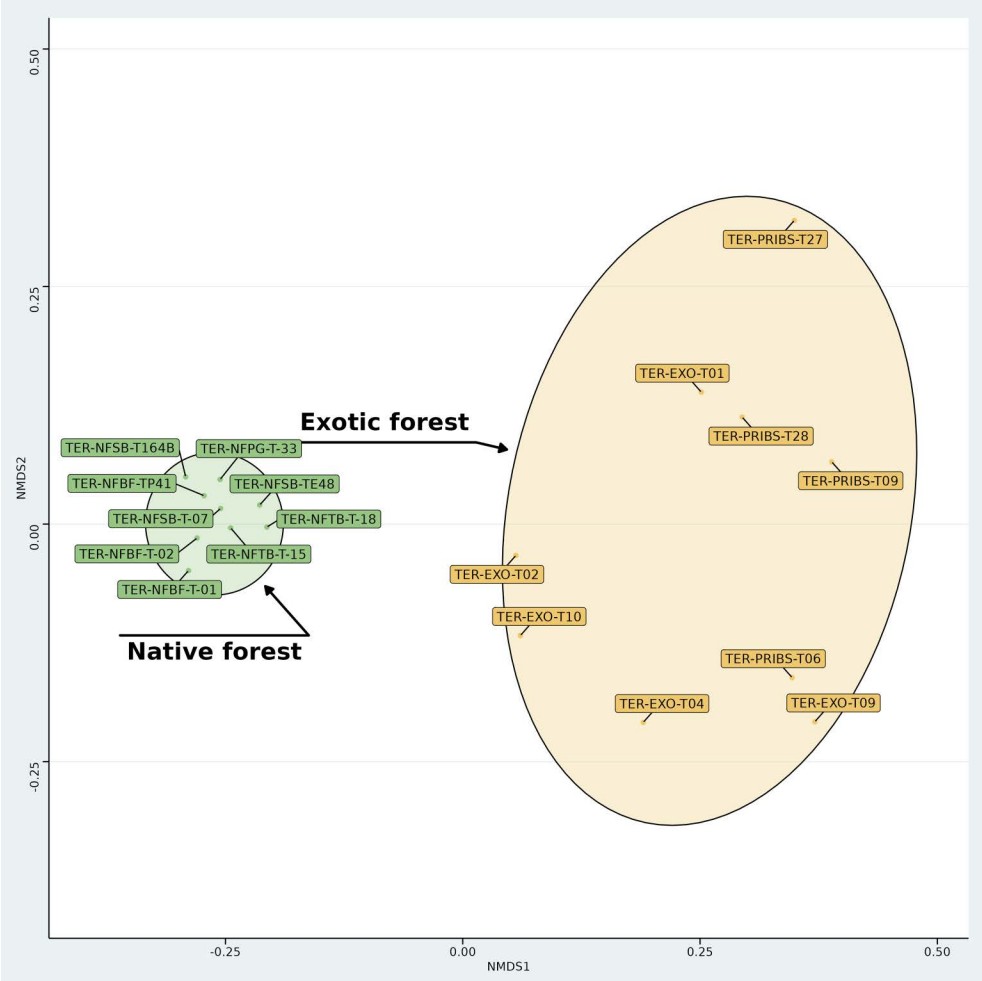

**Fig 3. NMDS Plot of Native vs. Exotic Forests on Terceira Island.** Non-metric Multidimensional Scaling (NMDS) plot showing the separation between native and exotic forest plots on Terceira Island, Azores, based on 49 variables. Each point represents an individual plot, with ellipses indicating the forest types (native (green) and exotic (yellow)). The clear spatial separation between the two ellipses reflects distinct ecological characteristics of native versus exotic forest structures, as supported by a significant ANOSIM result ($R = 0.9036$, $p < 0.001$).

## Discussion

We aimed to provide a detailed characterisation of the native and exotic woodlands on Terceira Island (Azores, Portugal). It was found that these forest types exhibited structural and compositional differences at different spatial scales. Furthermore, our study brings insights into the potential of forest structure as an indicator of ecological disturbance and invasion patterns, particularly in island ecosystems where native flora and fauna face unique vulnerabilities [22,73]. By analysing forest traits from the microhabitat to the landscape, our results reveal a multi-layered understanding of the ways in which both intrinsic ecological factors and external disturbances shape Azorean forests. The findings of this study highlight the necessity of examining forest structure at various spatial scales, as the differentiation between native and exotic forests exhibited significant variation depending on the spatial level under consideration.

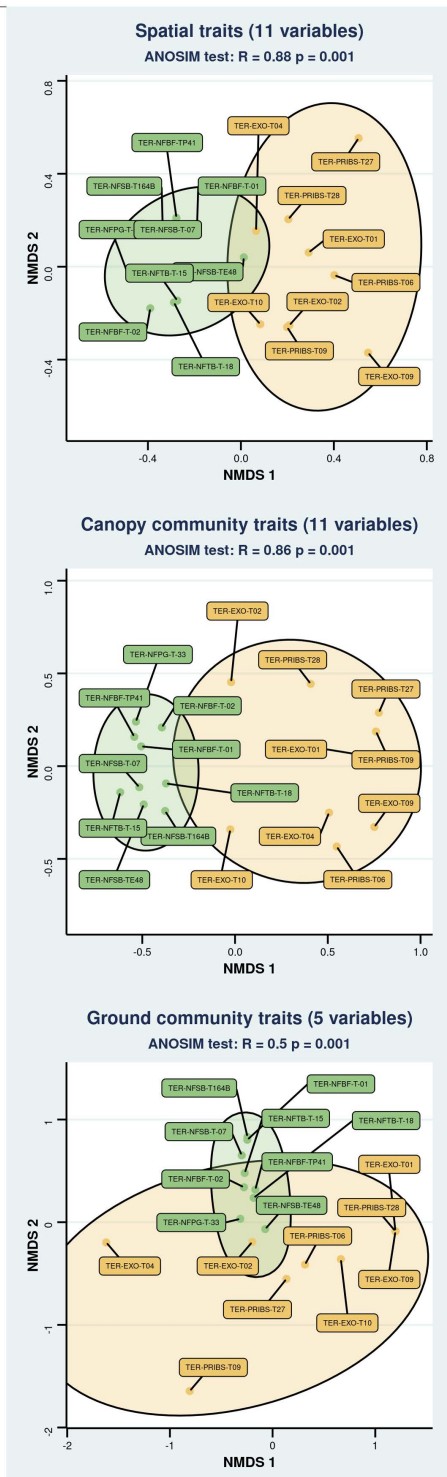

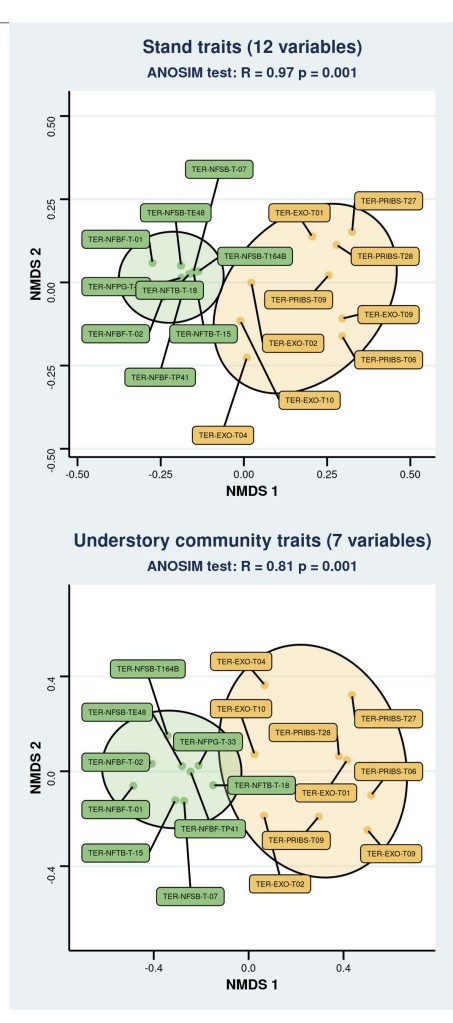

**Fig 4. NMDS Plots of Traits by Forest Type.** Non-metric Multidimensional Scaling (NMDS) plots for each trait considered. Each point represents an individual plot, with ellipses indicating the forest types (native (green) and exotic (yellow)). ANOSIM statistics and significance are mentioned for each graph.

**Table 2. Spearman rank correlation coefficients of variables significantly correlated with a NMDS axis for a given forest trait.**

| Trait | Variable | NMDS axis | Spearman's correlation coefficient (ρ) | p value |
|---|---|---|---|---|
| Spatial trait | naturalveg_prop | NMDS1 | −0,924 | <0,01 |
| Spatial trait | SRTM_elevation | NMDS1 | −0,748 | <0,01 |
| Spatial trait | slope_std | NMDS1 | −0,686 | 0,002 |
| Spatial trait | slope_max | NMDS1 | −0,498 | 0,035 |
| Spatial trait | pasture_prop | NMDS1 | 0,558 | 0,016 |
| Spatial trait | secondaryforest_prop | NMDS1 | 0,873 | <0,01 |
| Spatial trait | pasture_prop | NMDS2 | −0,704 | 0,001 |
| Spatial trait | slope_std | NMDS2 | 0,476 | 0,046 |
| Spatial trait | slope_max | NMDS2 | 0,517 | 0,028 |
| Spatial trait | slope_mean | NMDS2 | 0,723 | 0,001 |
| Spatial trait | slope_min | NMDS2 | 0,866 | <0,01 |
| Stand trait | mean_FD | NMDS1 | −0,909 | <0,01 |
| Stand trait | H0_VP | NMDS1 | −0,788 | <0,01 |
| Stand trait | basal_area_endemic | NMDS1 | −0,637 | 0,004 |
| Stand trait | basal_area_introduced | NMDS1 | 0,586 | 0,011 |
| Stand trait | ENL0D | NMDS1 | 0,688 | 0,002 |
| Stand trait | vertical_evenness | NMDS1 | 0,71 | 0,001 |
| Stand trait | basal_area_invasive | NMDS1 | 0,832 | <0,01 |
| Stand trait | ENL2D | NMDS1 | 0,917 | <0,01 |
| Stand trait | ENL1D | NMDS1 | 0,92 | <0,01 |
| Stand trait | foliage_height_diversity | NMDS1 | 0,92 | <0,01 |
| Stand trait | basal_area_native | NMDS2 | −0,714 | 0,001 |
| Canopy community trait | 2_endemic | NMDS1 | −0,935 | <0,01 |
| Canopy community trait | density_2_endemic | NMDS1 | −0,896 | <0,01 |
| Canopy community trait | density_2_introduced | NMDS1 | 0,51 | 0,031 |
| Canopy community trait | 2_introduced | NMDS1 | 0,6 | 0,009 |
| Canopy community trait | 2_invasive | NMDS1 | 0,776 | <0,01 |
| Canopy community trait | density_2_invasive | NMDS1 | 0,776 | <0,01 |
| Canopy community trait | density_2_native | NMDS2 | −0,725 | 0,001 |
| Canopy community trait | 2_native | NMDS2 | −0,634 | 0,005 |
| Canopy community trait | canopy_openness | NMDS2 | 0,882 | <0,01 |
| Understory community trait | 1_endemic | NMDS1 | −0,92 | <0,01 |
| Understory community trait | 1_native | NMDS1 | −0,82 | <0,01 |
| Understory community trait | UCI_mean | NMDS1 | −0,808 | <0,01 |
| Understory community trait | density_1_native | NMDS1 | 0,806 | <0,01 |
| Ground community trait | 0_endemic | NMDS1 | −0,53 | 0,024 |
| Ground community trait | density_0_introduced | NMDS2 | −0,546 | 0,019 |
| Ground community trait | 0_native | NMDS2 | 0,755 | <0,01 |
| Ground community trait | 0_endemic | NMDS2 | 0,902 | <0,01 |

## Landscape-level structure is a consequence of human accessibility and land-use

At the landscape level, spatial traits demonstrated a robust correlation between exotic forests and human-modified regions, whereas native forests exhibited a greater prevalence in remote, complex terrain. This pattern is common to all

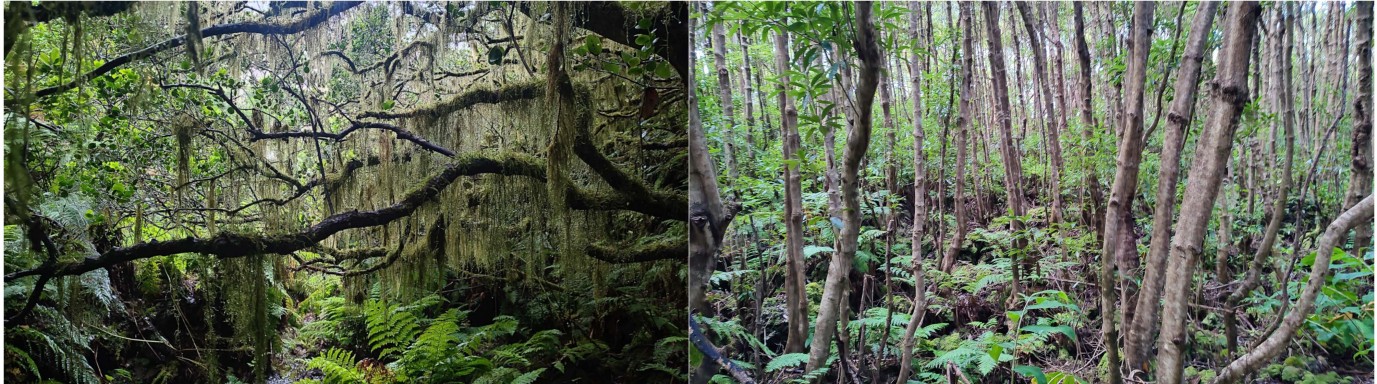

**Fig 5. Visual comparison of native and exotic woodlands.** Comparison of forest structures between native forest (left, TER-NFSB-T164B) and exotic woodlands (right, TER-PRIBS-T06) on Terceira Island. Visually, the native forest displays a complex vertical structure with diverse strata and species, while the exotic forest exhibits a reduced species diversity and overall structural complexity.

the Macaronesian islands [74], in which native forest is now restricted to more complex orographic terrains, being relics of past dramatic native forest clearing. This spatial distribution not only reflects the influence of human accessibility as a driver of ecosystem change [74], but also points to the role of historical land-use patterns, documented through both ecological studies [28] and historical accounts [39–41] which have progressively restricted native forests to higher elevations. Therefore, it is probable that the selected native forest plots, which are less accessible, have experienced a reduced number of direct disturbances over time, thereby preserving their structural complexity.

Moreover, the current fragments of Terceira's native forest are now under protection, part of integral natural reserves. Creation of protected areas is known to reduce the human impact (see for example [75,76]) but it must be based on a correct spatial distribution, otherwise the effectiveness of their protection is limited (see examples in [77–79]). However, all over the world, islands still face significant ecological challenges due to invasive plant species spread [22,80], and in the Azores several invasive plants are spreading in native forest [81,82]. In the current study, the selected nine plots of native forest in Terceira Island were chosen based on their high level of natural condition. However, despite the results obtained reflecting low levels of disturbance, some invasive plants are already spreading in some plots. For example, the spread of species such as *Pittosporum undulatum* (Australian cheesewood) or *Hedychium gardnerianum* (Ginger lily) has resulted in the decline of native flora and fauna in numerous native forest areas across several Azorean islands, particularly due to alterations in species interaction networks [83,84]. This impact is particularly concerning for the ecosystem of the Azores, which has evolved in isolation and where many plants and animals are not adapted to compete with aggressive newcomers [85]. Thus, while the forests are now protected, active management, like removing invasive species and restoring native vegetation, remains crucial to maintaining their health and resilience, which is being currently implemented under the scope of several LIFE projects (e.g., LIFE BEETLES [86]).

By contrast, exotic forests, which have been established in more easy-to-access areas, exhibit spatial traits that align with historical human activity and land-use change. Since settlement began approximately 500 years ago [41], human presence on Terceira Island has led to the intentional and unintentional introduction of non-native species, particularly in forests accessible for resource extraction and agriculture [87]. The enhanced accessibility has facilitated the widespread transformation of landscapes, where indigenous species were initially displaced through land-use change, and later further impacted by the expansion of exotic and invasive species into these disturbed areas [74]. Indeed, the non-indigenous tree species reported in these ecosystems, such as *Pittosporum undulatum* and *Cryptomeria japonica*, were introduced by humans for economic reasons [4,88]. In particular, *P. undulatum*, the dominant species in our exotic plot, was introduced

to the Azores as a windbreak to protect the fruit orchards [89]. When production decreased, the orchards were abandoned, and this species, with dispersal potential in the Azores [90], proliferated, creating a mosaic of secondary woodlands among pastures.

These large historical changes at the landscape scale can cascade down to finer structural changes at stand and microhabitat levels. Interestingly, some of these invaded areas still hold rare endemic arthropods, with relict populations surviving in these remnants [48].

## The structural complexity of forest stands is derived from different sources in native and exotic woodlands

Contrary to expectations, the Stand Structural Complexity Index (SSCI) was not identified as a statistically significant variable differentiating our native and exotic plots. Given its established use in numerous studies aimed at characterising structural differences linked to forest management [54,57], we would have expected this composite index to be a primary criterion for distinguishing between forest types.

It must be assumed that this is a consequence of the methodology employed in the construction of the index. The SSCI is defined as an exponential combination of the fractal dimension index (mean_FD, as defined by [91]) and the effective number of layers (ENL1D, as defined by [62]), which encompass the stand height for the purposes of providing a scaling component. However, mean_FD was identified as a relevant indicator for describing native stands, whereas ENL1D was associated with exotic woodlands. It can be inferred that SSCI will exhibit comparable values between the two forest types, limiting its use in distinguishing between native and exotic forests in the Azores. The similar values obtained for this composite index indicate that the structural complexity may have originated from different origins.

The intrinsic complexity of native stands may be attributed to the traits of the few endemic tree species (e.g., the Azorean endemic cedar *Juniperus brevifolia*), which exhibit greater branch ramifications, as well as the numerous non-vascular epiphytic species, resulting in a higher fractal dimension. Conversely, the structural complexity observed in exotic woodlands is likely to be the consequence of the mixture of invasive and introduced species and the overall greater height affecting the scaling-component of the SSCI index. Indeed, non-indigenous tree species were selected based on their capacity for rapid growth and their tendency to reach considerable heights, with the objective of utilising them as windbreaks. Consequently, exotic woodlands exhibit higher stand height, which allows for the development of a greater number of layers positively affecting the SSCI.

## The canopy of native forests shows greater structural and compositional homogeneity

The structure of forest canopies drives habitat specialisation and species richness [92,93]. Thus, a particular interest must be dedicated to this microhabitat.

The canopy of the native forests on Terceira exhibits a high degree of structural and compositional consistency. This homogeneity likely stems from the evolutionary adaptations of endemic species to environmental conditions above 500–600 m namely lower temperatures, higher winds and rainfall, and poor soils [28,94]. Consequently, their canopy architecture and slower growth rates contribute to a dense, closed canopy. Following the environmental filtering theory, that may resist the establishment of invasive species, at least with current environmental conditions, since the impact of climatic changes may reverse the current situation (see [95]). In contrast, the exotic forests, composed of various introduced species, display greater structural variation and openness, particularly in the canopy. This structural diversity could be a result of a human-induced species replacement, resulting in the introduction of species with differing growth habits and ecological strategies. It is also important to consider that the observed structural heterogeneity in exotic forests may partly reflect their younger age and earlier successional stage relative to native forests. However, given their dominance by fast-growing invasive species and history of colonizing previously disturbed areas, this heterogeneity also reflects ongoing ecological dynamics that may not necessarily lead to convergence with native forest structure over time.

Interestingly, canopy roughness did not emerge as a significant variable of differentiation. This finding could reflect the role of consistent environmental factors, such as wind, that shape the canopy structure of both forest types in similar ways, regardless of species composition. Alternatively, the similarity in canopy roughness might suggest that while exotic species have replaced the original native canopy, they exhibit similar spatial configurations, at least from above, possibly due to historical human selection of species suited to local environmental conditions.

## Structural vulnerability varies between microhabitats

Our study highlights that structural vulnerability to invasion differs across forest layers.

The ground layer appears to be the most susceptible to initial invasions, as evidenced by the high density of introduced species in exotic forest plots. Similar pattern was found in arthropods taxon which present a lower biotic integrity in the epigeal community of native forests [86]. This susceptibility can be attributed to several factors, including the greater availability of resources and the presence of vacant ecological niches in the ground layer, which provide fewer competitive barriers for incoming species. Furthermore, higher propagule pressure due to the proximity with other types of ecosystems [47,48], contributes to the rapid establishment and dominance of introduced species with superior effective spreading abilities in the ground layer of exotic forests.

The canopy layer of Azorean forests appears to be the least vulnerable microhabitat. In exotic forests, it is dominated by invasive species such as *Pittosporum undulatum*, which has competitive advantages [96,97]. In the absence of targeted restoration measures, this dominance limits the possibility of native species re-establishment, effectively locking the canopy into a state of low biodiversity and limited structural resilience. The species produces a substantial quantity of foliage, which creates a dense canopy that effectively excludes the growth of other understory species [96]. In contrast, the native forest canopy, distinguished by its closed structure and prevalence of a few complex endemic species, is inherently less susceptible to invasion. These climax communities, with their dense canopy architecture, provide few opportunities for new species to establish in the absence of significant natural or human-induced disturbances [97,98]. However, the inherent unsaturation of island forest communities – a condition where ecological niches remain unfilled – renders them susceptible to species packing following human introductions [20]. Any change in the structural complexity of these canopies, whether through disturbance or invasive dominance, would signify severe ecological shifts, altering habitat availability and impacting the broader ecosystem.

Beyond the dominance of *Pittosporum undulatum* in the canopy, our results reveal that exotic forests also harbour a diversity of introduced and invasive species in the lower layers. This pattern suggests a multi-invasion dynamic, where non-native species colonize distinct forest strata, potentially interacting and reinforcing structural and compositional shifts across vertical layers [99]. Such stratified invasions may accelerate ecological transformation by altering resource availability, microclimate, and species interactions at multiple levels within the forest, highlighting the complexity of managing biological invasions in island ecosystems.

Moreover, the understory layer in exotic forests shows greater resistance to invasion compared to the ground and canopy layers. This observation was also identified during the development of an Index of Biotic Integrity based on arthropod community composition [100]. It may be attributed to the competition with the high density of established native species or the existence of mutualistic relationships between indigenous species that reinforce the ecosystem's robustness. The understory may serve as a resilient microhabitat in the face of ecological disturbance, particularly the relative integrity of this microhabitat in the exotic plot may be attributed to the refuge function that this ecosystem provides for arthropods, as described by [48]. However, it is noteworthy that the structural complexity of this layer is already disturbed, as the exotic forest showed the lowest values for the Understory Complexity Index (UCI_mean).

## Temporal dynamics and conservation implications of forest change

Whilst it is acknowledged that a temporal discrepancy between vegetation surveys for native forest plots may potentially influence comparisons of species composition, it is considered that this impact is negligible in this context. The native

forests of the Azores, specifically those selected for this study, are situated in well-preserved and minimally disturbed areas within protected reserves [28]. These ecosystems are characterised by their ecological stability and slow successional dynamics [46], as they are dominated by long-lived endemic tree species [28,33] and exhibit limited exposure to human-induced pressures [74]. By contrast, exotic forest ecosystems are characterised by important dynamism, manifesting in both species composition and structural diversity, owing to the spatial heterogeneity in the surrounding environment [101,102]. Consequently, exotic forests are more prone to rapid change over short timescales, whereas native forests are expected to maintain a relatively consistent structural and compositional profile over the decade separating the surveys. Future long-term monitoring would be valuable to further validate these hypotheses.

These findings, while derived exclusively from Terceira Island, hold valuable implications for forest conservation across the Azores. Although extrapolation to other islands should be made with caution, the structural and compositional trends observed are likely to be applicable to similar forest types elsewhere in the archipelago as Terceira hosts some of the most representative and best-preserved remnants of native forest [103,104]. This ecosystem, with its intrinsic complexity and resistance to invasion, act as crucial biodiversity reservoirs [46].

Conservation efforts should prioritize protecting less accessible areas where native species have a competitive advantage. Conversely, restoration in exotic forests should aim to enhance structural resilience by controlling invasive species, particularly in the ground and canopy layers, while promoting native species to improve overall habitat quality. Conservation efforts should prioritize protecting less accessible areas where native species have a competitive advantage. Conversely, restoration in exotic forests should aim to enhance structural resilience by controlling invasive species, particularly in the ground and canopy layers, while promoting native species to improve overall habitat quality.

## Supporting information

**S1 Data. Pre-processed data used during analysis.**
(ZIP)

## Acknowledgments

We thank Fernando Pereira and Maria Teresa Fereira for their contribution during the vegetation survey in native forest. We would like to express our gratitude to Brais Casal, Juan Barrajo, Juan Carlos dos Santos, Martha Martínez and Robert Marian for their invaluable assistance during the plant inventory in the exotic plots. We would also like to acknowledge Dirk Böttger, Michael Unger and Paul Konrad Seidel for their help during the LiDAR fieldwork.

## Author contributions

**Conceptualization:** Sébastien Lhoumeau, Rui B. Elias, Paulo A. V. Borges.

**Data curation:** Sébastien Lhoumeau, Rui B. Elias, Dominik Seidel, Paulo A. V. Borges.

**Formal analysis:** Sébastien Lhoumeau, Dominik Seidel, Paulo A. V. Borges.

**Funding acquisition:** Rui B. Elias, Paulo A. V. Borges.

**Investigation:** Sébastien Lhoumeau, Rui B. Elias, Dominik Seidel, Rosalina Gabriel, Paulo A. V. Borges.

**Methodology:** Sébastien Lhoumeau, Rui B. Elias, Dominik Seidel, Paulo A. V. Borges.

**Project administration:** Rui B. Elias, Paulo A. V. Borges.

**Resources:** Rui B. Elias, Dominik Seidel, Paulo A. V. Borges.

**Supervision:** Rosalina Gabriel, Paulo A. V. Borges.

**Validation:** Sébastien Lhoumeau, Rui B. Elias, Rosalina Gabriel, Paulo A. V. Borges.

**Visualization:** Sébastien Lhoumeau.

**Writing – original draft:** Sébastien Lhoumeau, Rui B. Elias, Dominik Seidel, Rosalina Gabriel, Paulo A. V. Borges.

**Writing – review & editing:** Sébastien Lhoumeau, Rui B. Elias, Dominik Seidel, Paulo A. V. Borges.

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
