## [Decision Letter · Decision Letter 0]

13 Feb 2025

Dear Dr. Lhoumeau,

Thank you for submitting your manuscript to PLOS ONE. After careful consideration, we feel that it has merit but does not fully meet PLOS ONE’s publication criteria as it currently stands. Therefore, we invite you to submit a revised version of the manuscript that addresses the points raised during the review process.

I have received and had the opportunity to evaluate the reviews from three reviewers. While one reviewer appreciated the work conducted, the other two reviewers identified significant weaknesses in the methodological approach that must be addressed or clarified before proceeding with further evaluation.Therefore, I ask the authors to carefully consider the feedback provided by the reviewers when revising the manuscript, especially to clarify the most critical points.

We look forward to receiving your revised manuscript.

Kind regards,

Francesco Boscutti

Academic Editor

PLOS ONE

Journal Requirements:

“S.L. is funded by the Azorean Government Ph.D. grant numbers M3.1.a/F/012/2022. R.B.E, RG and P.A.V.B. are currently funded by the projects Azores DRCT Pluriannual Funding (M1.1.A/INFRAEST CIENT/001/2022) and FCT—Fundação para a Ciência e Tecnologia FCT[1]UIDB/00329/2020-2024 DOI 10.54499/UIDB/00329/2020 (Thematic Line 1—integrated ecological assessment of environmental change on biodiversity). D.S received no outside funding.”

“S.L. is funded by the Azorean Government Ph.D. grant numbers M3.1.a/F/012/2022.

R.B.E, RG and P.A.V.B. are currently funded by the projects Azores DRCT Pluriannual Funding (M1.1.A/INFRAEST CIENT/001/2022) and FCT—Fundação para a Ciência e Tecnologia FCT-UIDB/00329/2020-2024 DOI 10.54499/UIDB/00329/2020 (Thematic Line 1—integrated ecological assessment of environmental change on biodiversity).

D.S received no outside funding.”

“S.L. is funded by the Azorean Government Ph.D. grant numbers M3.1.a/F/012/2022.

R.B.E, RG and P.A.V.B. are currently funded by the projects Azores DRCT Pluriannual Funding (M1.1.A/INFRAEST CIENT/001/2022) and FCT—Fundação para a Ciência e Tecnologia FCT-UIDB/00329/2020-2024 DOI 10.54499/UIDB/00329/2020 (Thematic Line 1—integrated ecological assessment of environmental change on biodiversity).

D.S received no outside funding. “

5. In the online submission form, you indicated that “Data are available on request.”

Reviewers' comments:

Reviewer's Responses to Questions

**Comments to the Author**

1. Is the manuscript technically sound, and do the data support the conclusions?

Reviewer #1: Yes

Reviewer #2: No

Reviewer #3: Partly

2. Has the statistical analysis been performed appropriately and rigorously?

Reviewer #1: Yes

Reviewer #2: Yes

Reviewer #3: Yes

3. Have the authors made all data underlying the findings in their manuscript fully available?

Reviewer #1: No

Reviewer #2: No

Reviewer #3: Yes

4. Is the manuscript presented in an intelligible fashion and written in standard English?

Reviewer #1: Yes

Reviewer #2: No

Reviewer #3: No

Reviewer #1: The manuscript PONE-D-24-59937 (“Landscape to Microhabitat: Uncovering the Multiscale Complexity of Azorean Forests”) identify the structural and compositional differences between native and exotic woodlands on Terceira Island, Azores.

The manuscript is correct and it is good. The references is very good and the history very well explained and its justification. The work is very interesting.

The introduction and methodology is correct but it’s necessary some annotations (see below). With respect to the results, I see it as good, although is necessary basic information. Finally, in the discussion section, it was very well written.

Some annotations:

With respect to material and methods, the plots area (20 x 20 m) are enough? What determines the size of the plot? The maximum height of the trees?

The authors could considered include table 1 as supplementary material.

Figure 3 needs improvement.

It would be interesting for the authors to include a list of the species present in each plot and in each treatment (native and exotic woodlands) as supplementary material indicating, if possible, their origin species (native, endemic, introduced, invasive).

In addition, it would be good to include a short introduction to the results with a paragraph with the percentages of species present in each type of forest.

In the discussion, the authors indicates that were ten plot in native woodlands but really are nine (Second paragraph in “The characterization of landscapes is a consequence of human accessibility and land-use”)

Reviewer #2: Abstract … By analysing landscape, habitat, and microhabitat traits, we found that native forests are situated in less accessible terrains…. rephrase, the location of native forests is not a trait, or derived from the fact that they are native, just the opposite, they are the remains of a former native forest covering all the locations.

Abstract … homogeneous structural complexity derived from endemic vascular plant species…. I do not agree with "derived", it should be "and", adding enough exotic species or letting succession into the non-native forest will eventually result in complexity, complexity does not derive from endemicity (see comments on the Results).

Abstract ...exotic forests, which are situated in human-modified landscapes; Exotic woodlands are the human modified landscape, rephrase as "exotic forests, are the result of human actions, afforestation or introduction of invading tree plants.

Abstract ...The ground and canopy layers in exotic forests were more invaded by nonindigenous species, while the understory demonstrated greater resilience... very confusing statement, first an absolute need to resolve the difference between afforested areas and invaded areas, then also to differential use and actions of both, usually afforested areas have periodical cleaning of the understory?

Abstract ...Our findings suggest a temporal progression of forest change, from pristine native to disturbed exotic stands.; The authors have no follow up of the dynamics of the forest to support this statement, by contrary many references exist proving the Azores forest cover suffered a massive and historical change, the object of study are the results of the change, conclusions cannot be to prove the chosen model of exotic versus native exists, something that should be stated in the abstract as historical, but by contrary to explain the complexity, diversity etc of the 2 distinct “woodlands”

Abstract “...These insights highlight the necessity for scale-specific conservation, prioritising native forests in remote areas and controlling invasive species in exotic forests to enhance ecological resilience”; basically what the authors propose is to protect what already is protected not because is legally protected but because is as stated "in less accessible terrains" and to keep the exotic areas as exotic although this exotic are themselves invasive, this is an absurd

Introduction The structure of the introduction is inverted, the object of study should firstly be defined and then the first 3 paragraphs on the first page should follow.

Introduction The Azores, one of the most remote archipelagos in the North Atlantic Ocean, is made up of nine islands where forests contain a mix of native and exotic species, shaped by both natural processes and six centuries of human influence. (Elias et al. 2016). This or a modified version of this sentence should be the opening sentence of the Introduction, the phrase is however overly optimistic in the use of the concept of "Mosaic" or “Mix”, the Azores have certainly less than 5% of Native forest cover and therefore a Mosaic of 5% versus 95% is not a mosaic. This needs to be clearly expressed by the authors on the introduction. Several studies published by one of the authors (Elias) can be used to state and support this key point to the readers.

Introduction ...consist primarily of broad-leaved evergreen trees. There are according to Elias (2016), Dias et al. (2021), and others several types of vegetation including some that are not broad-lead. To characterize the broad-leaved evergreen trees as such without any further hint or name is insufficient; include also the references: Dias, Eduardo; Mendes, Cândida; Aguiar, Carlos (2021). Vegetação dos Açores. In Capelo, J.; Aguiar, C. (Eds.), Vegetação de Portugal. Lisboa: INCM, p. 155-179. ISBN 978-972-27-2879-9; Mucina, L.; Bültmann, H.; Dierßen, K.; Theurillat, J.-P.; Raus, T.; Carni, A.; Šumberová, K.; et al. "Vegetation of Europe: hierarchical floristic classification system of vascular plant, bryophyte, lichen, and algal communities". Applied Vegetation Science 19 (2016): 3-264; Fernández Prieto, J.C.; Aguiar, C.; Dias, E.; Fernández Prieto, J.A.; Aguiar, Carlos; Dias, Eduardo; FernÃ¡ndez Prieto, JosÃ© Antonio. "Description of some new syntaxa from the Azores archipelago". International Journal of Geobotanical Research 2 1 (2012): 111-116. http://dx.doi.org/10.5616/ijgr120007

Introduction ...many endemic taxa Characterize "Many", or compare with other similar forests, however we lack the sense/definition of forest and therefore a way to compare with other similar forests. There is reference in this introduction characterizing the types of forest and giving proper biogeographical framework.

Introduction ...As a result, the Azores’s landscape is now a mosaic of native and exotic forests, each playing a different ecological role… This is a bold statement and a conclusion, furthermore without references, see above the comment on the "Mosaic", do invading exotic plants play a role is this a political or a scientific statement?

Introduction ...Native forests are crucial for maintaining the Azorean island’s biodiversity, acting as refuges for endemic plants and animals (Lhoumeau and Borges 2023) Well… native forests are native in fact.

Introduction Finally, the vegetation data collected from surveys provides information on species composition and density across all layers. The data, gathered on Terceira Island, are used to address the following question: What are the defining characteristics of the vegetation structure of the various types of Azorean forests, with a particular focus on the native and exotic woodlands? By answering this question, this study aims to provide a detailed characterisation of Terceira's forests, offering insights into how both native and exotic forest types are integrated within the island's broader ecosystem. The reader finds out that after all the study concerns one island out of nine islands, that needs to be addressed on the proposed title "Landscape to Microhabitat: Uncovering the Multiscale Complexity of Terceira Island forests (Azores, Portugal)" but see further comments below.

Material and Methods. Study Area and Site selection The second part of the first paragraph needs to be moved to the introduction (to a new and so far lacking characterization of the Terceira forests and habitats)

Material and Methods. Study Area and Site selection The sentence "the third largest of the nine islands in the archipelago (figure 1)." should be moved to the first paragraph

Material and Methods. Study Area and Site selection The sentence "which are characterised by broadleaved evergreen trees and are home to a variety of endemic plant species" is conclusive and derives from the results themselves should deleted.

Material and Methods. Study Area and Site selection What is meant by "by exotic vegetation, primarily consisting of Pittosporum undulatum", Pittosporum undulatum is an invasive plant species very well known, documented etc. therefore 1) there should be detailed information on this species on the introduction, including reference to many publications concerning the Azores/Pittosporum undulatum question; 2) Pittosporum undulatum is an Invasive tree species and should be delt as invasive; 3) if there are no other dominant exotic species referred on the study (such as Cryptomeria japonica, the dominant exotic tree in the Azores... then the title of the manuscript should be further adapted/changed to fit this much more simple comparison. One Island and Native (even this can be explicit) versus Pittosporum invaded areas.

Material and Methods. Study Area and Site selection There is a lacking explanation on the 18 plots choice, it cannot just be because they are already in use in something else. Are they bioclimaticaly homogeneous? Are they similar in use, in historical use? Etc. A table with all the plots and as many characteristics as possible should be included, namely altitude, slope, topographic complexity, bioclimate, rainfall etc.......And an explanation on the choice of course.

Material and Methods. Study Area and Site selection What is meant by "Native vegetation" there can be native grasslands, native scrub, native bogs, native….. It does not mean native forest, the reader can be led to think that the "Native vegetation" on Figure 1, is the same as the Native forest on Figure 2, no true!

Material and Methods. Study Area and Site selection Why include here Figure 2? It looks like a result; in any case it should be moved and the plots clearly identified (meaning that the images on Figure 2 should correspond to 2 of the studied plots)

Material and Methods. Vegetation Survey The exotic forest plots were surveyed in the spring of 2023 within 20 x 20 meter plots, while the native forest plots were surveyed earlier, during the summer of 2012 and 2013, within larger 50 x 50 meter plots. This seems to be an insurmountable methodological problem with a very complex (impossible?) solution, a 10 year difference in the survey! Thats a decade, plants/trees do grow in 10 years. How can this be addressed when considering that other data such as "UAV canopy mapping" and "Terrestrial laser scanning" do not match the inventories? Even more taken into account that the authors state below that forests are homogeneous, were they 10 years ago? and then why should they still be homogeneous or even keep the same composition 10 years later?

Material and Methods. Vegetation Survey The exotic forest plots were surveyed in the spring of 2023 within 20 x 20 meter plots, while the native forest plots were surveyed earlier, during the summer of 2012 and 2013, within larger 50 x 50 meter plots. Why are the plots different in size?

Material and Methods. Vegetation Survey These surveys were designed following a standardised protocol (Borges et al. 2018). The protocol needs to be here described, but how can a protocol be applied in 2012 if it was published in 2018?

Material and Methods. Vegetation Survey The objective was to obtain comprehensive data on the composition of forest species. Eventually this statement should precede everything else.

Material and Methods. Vegetation Survey Due to the relative homogeneity of the native forests, no rarefaction process was applied to the data from these plots, allowing for the retention of all inventory data. This decision reflects the low variability in species composition within the native forest, ensuring a more accurate and robust comparison with exotic forest plots. This needs to be clearly explained, in any case it should belong to another part of the M & M, maybe data treatment? As it is corresponds to an empty sentence

Material and Methods. Vegetation Survey Species richness (H₀), representing the total number of species present in each plot, was recorded to evaluate biodiversity within the forest plots Therefore according to the authors species richness and biodiversity are the same concept? Does biodiversity increase with alien species? Is this a scientific statement or a political concession for managers of the Azorean landscape?

Material and Methods. Vegetation Survey Additionally, the basal area of trees with a diameter at breast height (DBH) greater than 10 cm was calculated in hectares. This variable provides an estimate of tree density and size, which is crucial for understanding forest biomass and its contribution to the overall structure of the forest. This sentence needs to be rewritten; the meaning is unclear.

Material and Methods. Vegetation Survey Another important metric, species density, was quantified by counting the number of shoots per species and per square meter, allowing for an assessment of vegetation density in different areas. More information about the protocol can be found in (Borges et al. 2018). Again at least partially the protocol should be detailed here, it was published in 2018 but the "native forest" plots surveyed 6 years before. This is confusing and not acceptable.

Material and Methods. Vegetation Survey These measurements were further classified in three distinct forest strata: the ground layer, the understory, and the canopy, offering a vertical profile of the forest structure. Furthermore, species were categorized into four groups—endemic, native, introduced, and invasive—resulting in a total of twelve variables, with each stratum and species category providing specific insights into the forest’s ecological composition and structure. I cannot understand this sentence, was DBH measured on the understory plants? How was density measured at the ground layer, just trees? Based on what did the authors segregate "introduced" and "invasive", a published list based on Azorean flora crossed with concepts like those published by Pyšek, Richardson and many others?

Material and Methods. Sites and lanscape characteristics For each of the 18 study sites on Terceira Island, detailed data on elevation and landscape characteristics were collected to better understand the environmental factors influencing forest structure. If the sites are not homogeneous in what concerns elevation and other geographical/ecological factors then how can they be compared?

Material and Methods. Sites and lanscape characteristics Furthermore, landscape heterogeneity was quantified in the surrounding area of each site. Land use data, obtained from the Azorean government, were processed and analysed in QGIS. A quadrat of side 500 meters was designed around all plots, and the proportion of land use was assessed per category. Data of the "Azorean government" should be of free access if so give an url on the text. This data are recent? Why not using the UAV to obtain the data? Is the classification of the "Azorean government" a clear match to the authors needs? See above the comment on Native vegetation and Native forest.

Material and Methods. Sites and landscape characteristics Table 1 might deserve a detailed comment, there are no invasive species on the ground layer

Material and Methods. Sites and landscape characteristics Table 1. Finally the reader understands that although there are references to arthropods and other types of non-plant diversity, this is a study of plant diversity, there are no other assessments being introduced on this manuscript. This a study of plants without reference to the actual plants, even less types of plant communities, very peculiar.

Material and Methods. Sites and landscape characteristics In fact what the authors seem to have studied are one peculiar type of native Forest (broad leaved although in the discussion Juniperus brevifolia is mentioned as dominant and this is not a broad leaf plant) and one type of exotic/invaded Forest (a Pittosporum dominated area, calling it forest another misleading concept)

Results The characterisation of landscapes is a consequence of human accessibility and land-use; this is what the authors decided to study not a conclusion of their study, in any case there no references to historical use of the Azorean forests on the introduction, a major issue since it clearly determines the framework of the study.

Results The structural complexity of forest stands is derived from different sources in native and exotic woodlands; what the authors are trying to say is that distinct “Natural” “successional” routes are being observed in distinct “forest”, obviously they are different, they are historically distinct, the easily accessed areas had since the XV century many distinct uses, please read historical papers! They were initially cut for wood, them burned and cultivated for wheat then…… plants with Pittosporum and invaded by Pittosporum (and many other exotic plants), therefore the sources are human (potentially infinite, where are invading plants coming from, how many a year/decade?) or based on the few refugia (a concept not explored on the text). Further more what the authors refer as native (without description detail etc.) are mature forests? Secondary forests on permanent ecological/successional stands derived from there “complex orography”, were they cut in the past (certainly they were although possibly not cultivated)

Results The intrinsic complexity of native stands may be attributed to the traits of the few endemic tree species (e.g. the Azorean endemic cedar Juniperus brevifolia), which exhibit greater branch ramifications, OR they can attributed to the fact that they or not the ecological equivalent of the Pittosporum stands, in order to solve this comundrum Native forests in areas not corresponding to refugia should be included, it might even occur that the dominant species in smooth areas to be distinct species.

Results The canopy of native forests shows greater structural and compositional homogeneity If they represent simplified versions of the Native Forest due to historical use and refugia position this is exactly what we all expect to happen.

Statistical complexity does not solve the problems found in the original data, to the end of the manuscript the biological descriptors are unclear, vascular plants? Vascular and Bryophytes? Arthropods?

Reviewer #3: Dear authors

After throughout analysis of the manuscript (MS), my general opinion is that the research question is interesting, but, despite analytical workflow being adequate, there are some flaws that may compromise the coherence of the MS. Also, the results are somewhat self-evident, for instance that natural/pristine forest is structurally more complex than artificial/exotic forest stands. Besides I would expect that some possible analysis of functional diversity between the two types of forest as the result of different history and drivers could have been addressed by author.

In concrete, I may point some of the flaws or aspects that are unclear I think should be addressed in a futures version of the MS.

1. The sample is too small to be representative. I understand that logistic effort might restrain the dimension of the sample buu, 18 plots hardly represent the ecological variability of the Azorean Forest.

2. It is very unclear the exact dates of sampling, because authors state that sampling has started in 2012’ and could mean that 13 plots were sampled at that time or thenceforth (when exactly?). Some sampling was made already in the summer of 2024. Therefore, part of the (already small) sample was made with an interval of maybe 12 years. This means that ecological succession may have led to composition and structure changes and some soil use change could have happened. I think you cannot compare plots with such a time span without accounting for time, which authors did not in their analysis.

Another unclear aspect is the discrepancy of dates of the survey of metrics: the method reference (Borges, 2018) is posterior to the putative dates of the survey (2012, 2013). I really have a difficulty with understanding what procedure was really used at the time.

3. The detailed technical description of devices and procedure of canopy survey by LIDAR is superfluous and cumbersome in the main text and should be either omitted or sent to supplementary materials.

4. I would like to see in a table how the parameters sampled in the surveys were converted to concrete trait variables and their scale values. The table presenting the 49 variables dos does not do this.

5. The sampling of plant species simply classified into ‘native’ and ‘exotic/invasive’ is a very poor floristic classification of plant species. I think it is hard to understand and overly simplistic.

6. The use of arbitrary surface for plots for the sake of homogeneity, not using minimal-area/rarefaction procedure is unjustified. Rarefaction can be used once for each forest type then the surface value used repeatedly.

7. The idea of indirect ordination by NMDS and then analysing driver correlates by Spearman correlation with axes is a good idea, although there are direct methods (canonical correlation or canonical correspondence analysis) that might have bee used instead. The real problem – I have serious difficulties in understanding – is that authors seem to have performed the analysis on a matrix with all the variables (!?). And then candidate explanatory variables were re-used (with Spearman correlation) to interpret NMDS axis? The procedure is not clear in the MS.

8. Please explain what is the ‘R statistic’. Is it Spearman correlation coefficient?

9. The procedure leading to variable groups is unclear. Whas it arbitrary/subjective?

10. Discussion is very elaborate and dissects each the main results. But, most of it does not derive from the MS results, so being arbitrary many times, as for instance, comments of flora diversity – when no flora was sampled, just ‘native/exotic’. S0, the discussion is absolutely exaggerated in its extension and mostly based on literature no the author´s own results. If not somewhat speculative or free-floating (not supported neither by results or literature).

Another example of irrelevance is the whole discussion of several-scale analysis, from micro-habitat to stand scale. Nothing in the sampling procedure separates different spatial scales. I recommend that, in future versions of MS, the discussion to be much shorter and sticking to discussion of results and only comparisons with relevant similar results or needed to interpretation. Otherwise, suddenly readers get a new review article inside the MS.

I should remind that results are, in great extent, very trivial ones.

Therefore, I recommend Major Revision of the MS.

Best regards.

A reviewer.

**Do you want your identity to be public for this peer review?** For information about this choice, including consent withdrawal, please see our Privacy Policy

Reviewer #1: No

Reviewer #2: No

Reviewer #3: No

---

## [Author Response · Author response to Decision Letter 1]

14 Mar 2025

Journal Requirements:

and

We have formatted the manuscript according to the guidelines provided in the PLOS ONE style templates, including the file naming conventions.

We have added the requested information regarding the permits in the Methods section.

“S.L. is funded by the Azorean Government Ph.D. grant numbers M3.1.a/F/012/2022. R.B.E, RG and P.A.V.B. are currently funded by the projects Azores DRCT Pluriannual Funding (M1.1.A/INFRAEST CIENT/001/2022) and FCT—Fundação para a Ciência e Tecnologia FCT[1]UIDB/00329/2020-2024 DOI 10.54499/UIDB/00329/2020 (Thematic Line 1—integrated ecological assessment of environmental change on biodiversity). D.S received no outside funding.”

“S.L. is funded by the Azorean Government Ph.D. grant numbers M3.1.a/F/012/2022.

R.B.E, RG and P.A.V.B. are currently funded by the projects Azores DRCT Pluriannual Funding (M1.1.A/INFRAEST CIENT/001/2022) and FCT—Fundação para a Ciência e Tecnologia FCT-UIDB/00329/2020-2024 DOI 10.54499/UIDB/00329/2020 (Thematic Line 1—integrated ecological assessment of environmental change on biodiversity).

D.S received no outside funding.”

We corrected the manuscript accordingly. The Funding Statement you provided is correct.

“S.L. is funded by the Azorean Government Ph.D. grant numbers M3.1.a/F/012/2022.

R.B.E, RG and P.A.V.B. are currently funded by the projects Azores DRCT Pluriannual Funding (M1.1.A/INFRAEST CIENT/001/2022) and FCT—Fundação para a Ciência e Tecnologia FCT-UIDB/00329/2020-2024 DOI 10.54499/UIDB/00329/2020 (Thematic Line 1—integrated ecological assessment of environmental change on biodiversity).

D.S received no outside funding. “

We added the statement that our funders did not have any role in the Funding Statement.

5. In the online submission form, you indicated that “Data are available on request.”

We have addressed the data availability requirements by adding all data used in the manuscript as supplementary materials.

Review Comments to the Author

Reviewer #1:

The manuscript PONE-D-24-59937 (“Landscape to Microhabitat: Uncovering the Multiscale Complexity of Azorean Forests”) identify the structural and compositional differences between native and exotic woodlands on Terceira Island, Azores.

The manuscript is correct and it is good. The references is very good and the history very well explained and its justification. The work is very interesting.

The introduction and methodology is correct but it’s necessary some annotations (see below). With respect to the results, I see it as good, although is necessary basic information. Finally, in the discussion section, it was very well written.

Some annotations:

With respect to material and methods, the plots area (20 x 20 m) are enough? What determines the size of the plot? The maximum height of the trees?

Thank you for your positive feedback on our manuscript. Regarding the plot size in the Materials and Methods section, the 20 x 20 m plots were determined based on accessibility and logistical considerations. Due to the small stature of Azorean forests, plots sizes usually range from 5x5 m to 15 x 15 m (e.g. Dias et al. 2011; Elias et al. 2011; Elias et al. 2016. Additionally, the empirical assessment of the homogeneity of the surroundings at the center of the plot justified the selection of this plot size.

The authors could considered include table 1 as supplementary material.

Figure 3 needs improvement.

Thank you for your suggestions. We have moved Table 1 to the supplementary material and added this mention in the beginning of the paragraph. Additionally, we have improved the visual quality of Figure 3 and incorporated some critical information from the former Table 1 into the figure for better clarity. Please note that figure 3 become figure 2 as the former figure 2 was moved to the results based on the comment of the second referee.

It would be interesting for the authors to include a list of the species present in each plot and in each treatment (native and exotic woodlands) as supplementary material indicating, if possible, their origin species (native, endemic, introduced, invasive).

Following the editor's guidelines, we have now provided all the data used in the analysis as supplementary materials. This includes a list of the species present in each plot and treatment (native and exotic woodlands), along with their colonization origin based on the updated information available in AZORESBIOPORTAL (native non-endemic, endemic, introduced, invasive).

In addition, it would be good to include a short introduction to the results with a paragraph with the percentages of species present in each type of forest.

We have added brief descriptive sentences at the beginning of the Results section as well as a table summarising our surveys.

In the discussion, the authors indicates that were ten plot in native woodlands but really are nine (Second paragraph in “The characterization of landscapes is a consequence of human accessibility and land-use”)

Thank you for pointing out the discrepancy. We have corrected the typo in the discussion section. Initially, ten plots were selected, but one plot was removed from the study due to the inability to obtain LiDAR data for it, resulting in a total of nine plots in the native woodlands.

Reviewer #2:

Abstract … By analysing landscape, habitat, and microhabitat traits, we found that native forests are situated in less accessible terrains…. rephrase, the location of native forests is not a trait, or derived from the fact that they are native, just the opposite, they are the remains of a former native forest covering all the locations.

We have revised the abstract to clarify that the current locations of native forests in less accessible terrains are not inherent traits of these forests.

Abstract … homogeneous structural complexity derived from endemic vascular plant species…. I do not agree with "derived", it should be "and", adding enough exotic species or letting succession into the non-native forest will eventually result in complexity, complexity does not derive from endemicity (see comments on the Results).

What we meant was that endemic trees, such as the dominant Juniperus brevifolia, exhibit a branching architecture that directly contributes to the structural complexity observed in native forests. To clarify this point, we have revised the text accordingly.

Abstract ...exotic forests, which are situated in human-modified landscapes; Exotic woodlands are the human modified landscape, rephrase as "exotic forests, are the result of human actions, afforestation or introduction of invading tree plants.

Thank you for your suggestion. The plots are in sites where there was afforestation followed by agricultural use that was latter abandoned, allowing the invasion of exotic tree species. We have rephrased the sentence to clarify that exotic forests result from human actions, including afforestation or the introduction of invasive tree species.

Abstract ...The ground and canopy layers in exotic forests were more invaded by nonindigenous species, while the understory demonstrated greater resilience... very confusing statement, first an absolute need to resolve the difference between afforested areas and invaded areas, then also to differential use and actions of both, usually afforested areas have periodical cleaning of the understory?

We acknowledge the importance of distinguishing between afforested and invaded areas. Based on our knowledge, nearly all exotic forest plots in this study were originally afforested for orchards or vineyards and later abandoned. Currently, there is no evidence of ongoing human intervention in these areas or their surroundings. However, we recognize the limitation that precise historical land-use data are not available. We also have clarified the point in the manuscript that the understory layer considered in this study is richer in terms of native species comparing to the ground and canopy layers.

Abstract ...Our findings suggest a temporal progression of forest change, from pristine native to disturbed exotic stands.; The authors have no follow up of the dynamics of the forest to support this statement, by contrary many references exist proving the Azores forest cover suffered a massive and historical change, the object of study are the results of the change, conclusions cannot be to prove the chosen model of exotic versus native exists, something that should be stated in the abstract as historical, but by contrary to explain the complexity, diversity etc of the 2 distinct “woodlands”

We agree and have rephrased the statement to remain more factual, emphasizing the historical context of forest change in the Azores and focusing on the structural and ecological characteristics of the two distinct woodland types.

Abstract “...These insights highlight the necessity for scale-specific conservation, prioritising native forests in remote areas and controlling invasive species in exotic forests to enhance ecological resilience”; basically what the authors propose is to protect what already is protected not because is legally protected but because is as stated "in less accessible terrains" and to keep the exotic areas as exotic although this exotic are themselves invasive, this is an absurd

We acknowledge the need to clarify our statement and have rephrased it to emphasize the importance of concrete conservation actions and continuous monitoring. While protected areas may be designated, their effective management and enforcement are essential to preserving native forests and controlling invasive species in exotic woodlands. In fact, we still see many areas in the archipelago (eg. In Flores, São Jorge and Faial) where the protection is only in the paper. Cattle is still moving around freely. This revision ensures a more action-oriented perspective in our conclusions.

Introduction The structure of the introduction is inverted, the object of study should firstly be defined and then the first 3 paragraphs on the first page should follow.

Thank you for your suggestion. However, we have chosen to maintain the current structure of the introduction, as it follows a standard scientific writing approach—first presenting the broader context of the study before defining its specific objectives. We believe this approach enhances the clarity of the narrative, providing a coherent and logical progression of ideas that guides the reader from general background information to the precise aims of the research.

Introduction The Azores, one of the most remote archipelagos in the North Atlantic Ocean, is made up of nine islands where forests contain a mix of native and exotic species, shaped by both natural processes and six centuries of human influence. (Elias et al. 2016). This or a modified version of this sentence should be the opening sentence of the Introduction, the phrase is however overly optimistic in the use of the concept of "Mosaic" or “Mix”, the Azores have certainly less than 5% of Native forest cover and therefore a Mosaic of 5% versus 95% is not a mosaic. This needs to be clearly expressed by the authors on the introduction. Several studies published by one of the authors (Elias) can be used to state and support this key point to the readers.

We have rephrased "mosaic" to "patches" to more accurately reflect the limited extent of native forest cover in the Azores. Additionally, we have clarified this point in the introduction, citing relevant studies, to ensure readers understand the significant imbalance between native and exotic forest areas.

Introduction ...consist primarily of broad-leaved evergreen trees. There are according to Elias (2016), Dias et al. (2021), and others several types of vegetation including some that are not broad-lead. To characterize the broad-leaved evergreen trees as such without any further hint or name is insufficient; include also the references: Dias, Eduardo; Mendes, Cândida; Aguiar, Carlos (2021). Vegetação dos Açores. In Capelo, J.; Aguiar, C. (Eds.), Vegetação de Portugal. Lisboa: INCM, p. 155-179. ISBN 978-972-27-2879-9; Mucina, L.; Bültmann, H.; Dierßen, K.; Theurillat, J.-P.; Raus, T.; Carni, A.; Šumberová, K.; et al. "Vegetation of Europe: hierarchical floristic classification system of vascular plant, bryophyte, lichen, and algal communities". Applied Vegetation Science 19 (2016): 3-264; Fernández Prieto, J.C.; Aguiar, C.; Dias, E.; Fernández Prieto, J.A.; Aguiar, Carlos; Dias, Eduardo; FernÃ¡ndez Prieto, JosÃ© Antonio. "Description of some new syntaxa from the Azores archipelago". International Journal of Geobotanical Research 2 1 (2012): 111-116. http://dx.doi.org/10.5616/ijgr120007

We agree that a more detailed characterization of the broad-leaved evergreen trees in the Azores is important. We have included peer-reviewed relevant references to clarify the diversity of native vegetation types in the archipelago. Additionally, we have provided a clear definition of the native woodlands considered in this study, which aims to provide a broad overview of the structural complexity of the main woodlands and their potential determinants.

Introduction ...many endemic taxa Characterize "Many", or compare with other similar forests, however we lack the sense/definition of forest and therefore a way to compare with other similar forests. There is reference in this introduction characterizing the types of forest and giving proper biogeographical framework.

We cited several references that detail the endemic species composition, including recent and updated checklists. To address your comment, we have now included a clear definition of the native vegetation type considered in this study in the Materials and Methods secti

---

## [Decision Letter · Decision Letter 1]

12 Apr 2025

Dear Dr. Lhoumeau,

Thank you for submitting your manuscript to PLOS ONE. After careful consideration, we feel that it has merit but does not fully meet PLOS ONE’s publication criteria as it currently stands. Therefore, we invite you to submit a revised version of the manuscript that addresses the points raised during the review process.

We look forward to receiving your revised manuscript.

Kind regards,

Francesco Boscutti

Academic Editor

PLOS ONE

Journal Requirements:

Reviewers' comments:

Reviewer's Responses to Questions

**Comments to the Author**

Reviewer #1: All comments have been addressed

Reviewer #2: (No Response)

Reviewer #3: All comments have been addressed

2. Is the manuscript technically sound, and do the data support the conclusions?

Reviewer #1: Yes

Reviewer #2: Partly

Reviewer #3: Yes

3. Has the statistical analysis been performed appropriately and rigorously?

Reviewer #1: Yes

Reviewer #2: Yes

Reviewer #3: Yes

4. Have the authors made all data underlying the findings in their manuscript fully available?

Reviewer #1: Yes

Reviewer #2: Yes

Reviewer #3: Yes

5. Is the manuscript presented in an intelligible fashion and written in standard English?

Reviewer #1: Yes

Reviewer #2: Yes

Reviewer #3: (No Response)

Reviewer #1: The work is very interesting and has improved a lot with respect to the previous version. All the changes proposed by the authors of this paper have been done correctly and those who have not agreed, the authors have justified it very well.

Reviewer #2: I keep my comment on the title: The reader finds out that after all the study concerns one island out of nine islands, that needs to be addressed on the proposed title "Landscape to Microhabitat: Uncovering the Multiscale Complexity of Terceira Island forests (Azores, Portugal)". The authors have no data on the Azores just for 1/9 of the islands.

Abstract

By analysing landscape(…) we found that less accessible terrains are covered by the remnants of native forests. (…) plant species.

The authors are again stating that “they found” instead rephrase as:

The authors state: However, we recognize the limitation that precise historical land-use data are not available.

In fact historical data on the Azores do exist, it’s not of course digital cartography from the XV/XVI centuries but several descriptions, or recent publications that synthetise these ancient documents (see e.g. Dias, on the book: Açores e Madeira - A Floresta das Ilhas. Pp. 255-296. Público, Comunicação Social SA, Fundação Luso Americana para o Desenvolvimento e Liga para a Protecção da Natureza. ISBN: 978-989-619-103-0)

Abstract

Comment by the authors: While protected areas may be designated, their effective management and enforcement are essential to preserving native forests and controlling invasive species in exotic woodlands. In fact, we still see many areas in the archipelago (eg. In Flores, São Jorge and Faial) where the protection is only in the paper. Cattle is still moving around freely.

Reply: I understand the comment and the commitment but there cannot be implicit conclusions in a paper, or the results show what the authors state on the comment or my comments, based solely on the authors data, are still pertinent. (see final remark)

The statement “These insights emphasize (…) the preservation of native forests and the control of invasive species in exotic woodlands.” Is therefore hard to accept as it is, the authors data are not on the management, or protected versus unprotected areas, from the authors data these conclusions cannot be drawn.

Introduction

The authors state:

However human activity significantly altered, with large areas being replaced by exotic species such as Cryptomeria japonica (L. f.) D.Don, 1841 and Pittosporum undulatum Vent., introduced for commercial forestry [29]. As a result, the Azores’s landscape has several patches of native and exotic woodlands in a matrix of agricultural fields [32,36], each with a different ecological importance. Native vegetation are crucial for maintaining the Azorean island’s biodiversity, acting as refuges for endemic plants and animals [37].

Delete “1841”

Later in the comments the authors refer to 33 % for Pittosporum, 31 for % Cryptomeria, since both pastures, vineyards and many other human uses of the landscape are still missing I guess that “significantly altered” means close to extinction? The authors need here to introduce values/numbers on the % of native vegetation remains and native forest (and cite the sources).

Introduction ...consist primarily of broad-leaved evergreen trees. There are according to Elias (2016), Dias et al. (2021), and others several types of vegetation including some that are not broad-lead. To characterize the broad-leaved evergreen trees as such without any further hint or name is insufficient; include also the references: Dias, et al (2021). Vegetação dos Açores. In Capelo, J.; Aguiar, C. (Eds.), Vegetação de Portugal. Lisboa: INCM, p; Mucina, et al. "Vegetation of Europe: hierarchical floristic classification system of vascular plant, bryophyte, lichen, and algal communities". Applied Vegetation Science 19 (2016): 3-264; Fernández Prieto, et al. "Description of some new syntaxa from the Azores archipelago". International Journal of Geobotanical Research 2 1 (2012): 111-116. http://dx.doi.org/10.5616/ijgr120007

The references included in my first review need to be included, there can be no cancellation on authors views or scientific approaches to vegetation study.

The authors state: “each with a different ecological importance” this is unacceptable for island vegetation/plant diversity since none of the human introduced uses is ecological important, all being ecological destructive. The statement says a lot on authors views on conservation. It needs to be deleted.

Introduction

Although the authors revised the use of “mosaic” this part of the introduction still needs to be improved using numerical data on types of cover.

Material and Methods.

1st paragraph

“…vegetation which were dominated” rephrase as “vegetation which are dominated”

What’s the meaning of “primarily consisting” means that are sites not dominated by Pittosporum?

Vegetation survey

“The objective was to obtain comprehensive data on the composition of forest species.” Rephrase objectives do not fit M&M

Material and Methods.

Cryptomeria is not the dominate exotic species. Pittosporum dominated areas occupy 33% of forest surface in the Azores. Production forests (mostly, but not only (!), of Cryptomeria) occupy 31%. This is according to the «State of the Environment Report of 2023» (Azorean Regional Government).

I accept that a further change on the title (apart from the geographical restriction) is not need, but use this data (numerical data) where required (see comment above)

Material and Methods. Study Area and Site selection

The authors in their reply refer to a table, I could not find the reference in the text.

Material and Methods.

Although the authors have improved the text concerning plots and time, both the difference in size and the 10 years difference need to be included in the discussion.

Original comment: Even more taken into account that the authors state below that forests are homogeneous, were they 10 years ago? and then why should they still be homogeneous or even keep the same composition 10 years later?

I agree that the authors have the needed information, the question is where do readers get that information? And there is no reference to that in the text, it needs to be included with references preferably published in peer-reviewed publications (as stated above by the authors)

The authors further reply: Conducting new surveys could risk altering the forest structure due to trampling, which is particularly important since the focus of this study is on the structural complexity of the forests.

This seems to be a very poor, and not included, explanation.

The authors further reply: Similarly, (…) we believe the 10-year difference does not undermine this objective.

In my view this weakness on the methodological approach cannot be solved just by answering to the reviewers using private and unpublished information.

The authors have two choices to publish the information in the paper or to cite another source.

Material and Methods. Vegetation Survey

Original comment: The exotic forest plots were surveyed in the spring of 2023 within 20 x 20 (…). Why are the plots different in size?

Reply by the authors: The larger 50 x 50 meter plots for the native forests were chosen during the design of the ISLANDBIODIV project and we use the data gathered. The 20 x 20 meter plots in the exotic forests were selected based on forest height, accessibility and logistical considerations. Furthermore, the homogeneity of the plots supported the selection of this area to represent the forest structure being studied.

Homogeneity is a type of data, not shown in the paper, just assumed in the reply, if the authors can show or produce evidence of this then the reply should be transferred into the text. Otherwise, its complex to accept.

Material and Methods. Vegetation Survey

Original comment: The protocol needs to be here described, but how can a protocol be applied in 2012 if it was published in 2018?

Authors reply: (…). The vegetation survey protocol was initially designed during earlier projects, notably ISLANDBIODIV and MOVECLIM, and was implemented in 2012 (…), leading to a delay in its official release.

I still don’t see the reply to “The protocol needs to be here described”

So, the readers are still missing a crucial point on methods used, describe the protocol even if just in a shorten version, was it adapted in any of its parts?

Material and Methods. Vegetation Survey

Original comment: Based on what did the authors segregate "introduced" and "invasive", a published list based on Azorean flora crossed with concepts like those published by Pyšek, Richardson and many others?

Authors reply: (…) The classification of species into "endemic," "native," "introduced," and "invasive" is based on Borges PAV, Costa A, (…). Principia, Parede. It compiles data from various sources and more recent updates available in AZORES BIOPORTAL

However, the original 2010 publication, for plants does not refer to naturalized versus invasive, just “Natu”, naturalized, please clarify this point, i.e. the distinction between introduced and invasive.

Further comments:

On the results the authors state: “high level of pristine composition when considering the endemic tree species.”

This sentence seems to contradict the absence of data on pristine forests (see above), in fact it implies that there is a model of pristine vegetation, is there?

Discussion

Is lacking a thorough discussion of possible drawbacks of methodological problems and adaptations, and in my view is clearly over-extended.

1st paragraph: “Furthermore, (…) invasion patterns” it seems a circular though/statement, if invaded i.e. Pittosporum dominated sites have a distinct structure then structure is an indicator of the dominance of Pittosporum….why should anyone use structure if the “indicator” is at plain sight?

1st Paragraph “The findings of this study highlight the necessity of examining forest structure at various spatial scales, as the differentiation between native and exotic forests exhibited significant variation depending on the spatial level under consideration.”

Consider discussing your results on a multi-invasion process, i.e. where Pittosporum is the dominant invading phanerophyte but other invading plants at distinct forest strata are also paving their way into the landscape.

2nd Paragraph: “concerning for the delicate ecosystem of the Azores” well very threatened, reduced, relictual?

4th Paragraph: “human presence on Terceira Island has led to the intentional and unintentional introduction of non-native species, particularly in forests accessible for resource extraction and agriculture.” If this this is a discussion, please insert a citation for this statement.

5th Paragraph: “The enhanced accessibility (...) replacement of indigenous species with exotic ones” replacement would be understood by most of the readers as a direct replacement where one species (invasive) outcompeted a native/endemic. This was not the case. In the Azores vegetation was destroyed and used and later invaded, invasive species were introduced mostly in “empty” landscapes.

5th Paragraph: “By contrast, exotic forests, which have been established in more easy-to-access areas, exhibit spatial traits that align with historical human activity and land-use change.” Easy to access areas were more easily used by humans (agriculture etc) and therefore invaded because they were empty…

On the “The canopy ...”

Complexity is here assumed in a static view but if native forest is older (even very optimistically stated as close to pristine…) and exotic younger then heterogeneity would be expected do to time alone, i.e. the exotic forests are still far from “mature” where “natives” are mature forests. Please discuss.

On the “temporal dynamics”

“The findings indicate a historical trajectory (...) from pristine native forest ecosystems to increasingly disturbed exotic stands.” This not the scope of this publication that does not include historical data, even less on pristine forest. Delete the sentence.

“Conservation efforts should prioritize protecting less accessible areas where native species have a competitive advantage. Conversely, restoration in exotic forests should aim to enhance structural resilience by controlling invasive species, particularly in the ground and canopy layers, while promoting native species to improve overall habitat quality”.

Results prove clearly that conservation efforts should prioritize invaded areas, promoting their conversion to a higher degree of “nativeness” and not to preserve what is already protected as stated by the authors as close to pristine.

Reviewer #3: (No Response)

**Do you want your identity to be public for this peer review?** For information about this choice, including consent withdrawal, please see our Privacy Policy

Reviewer #1: No

Reviewer #2: No

Reviewer #3: No

---

## [Author Response · Author response to Decision Letter 2]

29 Apr 2025

I keep my comment on the title: The reader finds out that after all the study concerns one island out of nine islands, that needs to be addressed on the proposed title "Landscape to Microhabitat: Uncovering the Multiscale Complexity of Terceira Island forests (Azores, Portugal)". The authors have no data on the Azores just for 1/9 of the islands.

We agree that the title should accurately reflect the geographic extent of the study, which is limited to Terceira Island. Accordingly, we have revised the title to:

“Landscape to Microhabitat: Uncovering the Multiscale Complexity of Native and Exotic Forests on Terceira Island (Azores, Portugal)”

While our study focuses exclusively on Terceira Island, we emphasize in the manuscript that the findings are likely applicable to other Azorean islands. This is because Terceira hosts the most representative and best-preserved remnants of several types of native forest, as well as typical secondary and exotic forest types found throughout the archipelago. Given this ecological representativeness, we consider Terceira an appropriate model for broader ecological inferences across the Azores. We have now clarified this point more explicitly in the manuscript to ensure that readers properly understand the study’s wider relevance and contextual framework.

Abstract

By analysing landscape(…) we found that less accessible terrains are covered by the remnants of native forests. (…) plant species.

The authors are again stating that “they found” instead rephrase as:

We have revised the sentence in the abstract to avoid the assertive phrasing “we found.” The revised sentence now reads:

“Based on landscape, habitat, and microhabitat analyses, remnants of native forests appeared to be associated with less accessible terrains.”

The authors state: However, we recognize the limitation that precise historical land-use data are not available.

In fact historical data on the Azores do exist, it’s not of course digital cartography from the XV/XVI centuries but several descriptions, or recent publications that synthetise these ancient documents (see e.g. Dias, on the book: Açores e Madeira - A Floresta das Ilhas. Pp. 255-296. Público, Comunicação Social SA, Fundação Luso Americana para o Desenvolvimento e Liga para a Protecção da Natureza. ISBN: 978-989-619-103-0)

We agree that while detailed historical cartography may not be available, valuable descriptive and synthesized sources do exist. In response, we now have revised the manuscript to include previously missing relevant references that discuss historical ecosystem distribution in the Azores.

Abstract

Comment by the authors: While protected areas may be designated, their effective management and enforcement are essential to preserving native forests and controlling invasive species in exotic woodlands. In fact, we still see many areas in the archipelago (eg. In Flores, São Jorge and Faial) where the protection is only in the paper. Cattle is still moving around freely.

Reply: I understand the comment and the commitment but there cannot be implicit conclusions in a paper, or the results show what the authors state on the comment or my comments, based solely on the authors data, are still pertinent. (see final remark)

The statement “These insights emphasize (…) the preservation of native forests and the control of invasive species in exotic woodlands.” Is therefore hard to accept as it is, the authors data are not on the management, or protected versus unprotected areas, from the authors data these conclusions cannot be drawn.

We agree that our study does not include quantitative data on the effectiveness of protected area management or enforcement in the Azores as it was not our objective. To address this, we have revised the statement to remain within the scope of our findings. The sentence now reads:

“These insights emphasize the importance of long-term monitoring and structural assessments in informing conservation efforts aimed at preserving native forests and managing invasive species in exotic woodlands.”

While our field experience has exposed us to signs of inconsistent enforcement (e.g., cattle presence in some designated areas), such observations can be anecdotal and not sufficient to support empirical conclusions. We have therefore removed any implication that our data directly assess the effectiveness of protected area management.

Introduction

The authors state:

However human activity significantly altered, with large areas being replaced by exotic species such as Cryptomeria japonica (L. f.) D.Don, 1841 and Pittosporum undulatum Vent., introduced for commercial forestry [29]. As a result, the Azores’s landscape has several patches of native and exotic woodlands in a matrix of agricultural fields [32,36], each with a different ecological importance. Native vegetation are crucial for maintaining the Azorean island’s biodiversity, acting as refuges for endemic plants and animals [37].

Delete “1841”

The date “1841” has been removed from the species citation for Cryptomeria japonica, and the sentence has been corrected accordingly.

Later in the comments the authors refer to 33 % for Pittosporum, 31 for % Cryptomeria, since both pastures, vineyards and many other human uses of the landscape are still missing I guess that “significantly altered” means close to extinction? The authors need here to introduce values/numbers on the % of native vegetation remains and native forest (and cite the sources).

We have incorporated references that provide data on land-use changes and the current extent of native forest areas across the archipelago.

Specifically, we now cite Castanho et al. (2021), which offers a detailed analysis of land-use trends in the Azores between 1990 and 2018, and Triantis et al. (2010), which includes estimates of the remaining native forest cover and the historical context of habitat loss. These sources support the statement regarding significant and dramatic alteration of native vegetation, which refers not only to the replacement by Cryptomeria japonica and Pittosporum undulatum but also to broader land-use conversion for agriculture, pastures, and other human activities.

Introduction ...consist primarily of broad-leaved evergreen trees. There are according to Elias (2016), Dias et al. (2021), and others several types of vegetation including some that are not broad-lead. To characterize the broad-leaved evergreen trees as such without any further hint or name is insufficient; include also the references: Dias, et al (2021). Vegetação dos Açores. In Capelo, J.; Aguiar, C. (Eds.), Vegetação de Portugal. Lisboa: INCM, p; Mucina, et al. "Vegetation of Europe: hierarchical floristic classification system of vascular plant, bryophyte, lichen, and algal communities". Applied Vegetation Science 19 (2016): 3-264; Fernández Prieto, et al. "Description of some new syntaxa from the Azores archipelago". International Journal of Geobotanical Research 2 1 (2012): 111-116. http://dx.doi.org/10.5616/ijgr120007

The references included in my first review need to be included, there can be no cancellation on authors views or scientific approaches to vegetation study.

Many thanks for remembering those references, we have now included the suggested sources—Dias et al. (2021), Mucina et al. (2016), and Fernández Prieto et al. (2012)—alongside the existing citations. Altogether, these references provide a global view of the diversity of vegetation types in the Azores, beyond the general characterisation of broad-leaved evergreen trees.

The authors state: “each with a different ecological importance” this is unacceptable for island vegetation/plant diversity since none of the human introduced uses is ecological important, all being ecological destructive. The statement says a lot on authors views on conservation. It needs to be deleted.

While we fully agree that native vegetation holds the highest ecological value and should be prioritized in conservation, we would like to clarify that our intent was not to equate the ecological importance of native and human-modified ecosystems, but rather to acknowledge the ecological roles that even altered landscapes can occasionally play.

To support this nuance, we refer to the study by Tsafack et al. (2021), which demonstrates that small patches of exotic forest—although human-modified—can serve as refuges for rare endemic arthropods. We have therefore revised the text to avoid any misinterpretation and to clarify that the ecological value of these systems is context-dependent and generally lower than that of native forests, but not necessarily negligible.

The revised sentence now reads:

“As a result, the Azores’s landscape has several patches of native and exotic woodlands in a matrix of agricultural fields [32,42–44], each contributing differently to the current ecological dynamics of the island.”

We hope this rephrasing addresses the concern while still reflecting the current complex ecological realities observed in the field.

Introduction

Although the authors revised the use of “mosaic” this part of the introduction still needs to be improved using numerical data on types of cover.

Following your previous comment, wee have enhanced the list of references in the Introduction. They now include sources that provide quantitative data and detailed descriptions of land-use types in the Azores. In particular, the study by Castanho et al. (2021) and Triantis et al (2010) offer comprehensive data on land-cover categories and their evolution over recent decades and has now been cited to support our characterization of the landscape mosaic. We chose not to duplicate these statistics directly in the Introduction, as such information can be found in the cited sources, allowing interested readers to consult the detail.

Material and Methods.

1st paragraph

“…vegetation which were dominated” rephrase as “vegetation which are dominated”

We have corrected the verb tense as recommended. The sentence now reads:

“…vegetation which are dominated…”

What’s the meaning of “primarily consisting” means that are sites not dominated by Pittosporum?

The expression “primarily consisting” was intended to indicate that the dominant woody species in the exotic forest plots is Pittosporum undulatum. To avoid ambiguity, we have revised the sentence to read:

“The remaining nine sites were situated in areas dominated by the invasive species Pittosporum undulatum.”

Vegetation survey

“The objective was to obtain comprehensive data on the composition of forest species.” Rephrase objectives do not fit M&M

We agree that objectives are generally not stated in the Materials and Methods section. However, this sentence refers specifically to the aim of the vegetation survey, not the overall study. To avoid confusion, we have rephrased the sentence for clarity and alignment with standard scientific writing:

“This survey aimed to collect detailed data on forest species composition across the study plots.”

Material and Methods.

Cryptomeria is not the dominate exotic species. Pittosporum dominated areas occupy 33% of forest surface in the Azores. Production forests (mostly, but not only (!), of Cryptomeria) occupy 31%. This is according to the «State of the Environment Report of 2023» (Azorean Regional Government).

I accept that a further change on the title (apart from the geographical restriction) is not need, but use this data (numerical data) where required (see comment above)

Following your previous comments, we confirm that this observation has been addressed in the revised manuscript.

Material and Methods. Study Area and Site selection

The authors in their reply refer to a table, I could not find the reference in the text.

The table referred to is included in the supplementary material, and we have now added an explicit reference to it in the main text to ensure clarity and accessibility for the reader.

Material and Methods.

Although the authors have improved the text concerning plots and time, both the difference in size and the 10 years difference need to be included in the discussion.

Original comment: Even more taken into account that the authors state below that forests are homogeneous, were they 10 years ago? and then why should they still be homogeneous or even keep the same composition 10 years later?

I agree that the authors have the needed information, the question is where do readers get that information? And there is no reference to that in the text, it needs to be included with references preferably published in peer-reviewed publications (as stated above by the authors)

The authors further reply: Conducting new surveys could risk altering the forest structure due to trampling, which is particularly important since the focus of this study is on the structural complexity of the forests.

This seems to be a very poor, and not included, explanation.

The authors further reply: Similarly, (…) we believe the 10-year difference does not undermine this objective.

In my view this weakness on the methodological approach cannot be solved just by answering to the reviewers using private and unpublished information.

The authors have two choices to publish the information in the paper or to cite another source.

We have added a paragraph in the Discussion section addressing the limitations related to the time gap and the difference in plot size between the native and exotic vegetation surveys. We acknowledge that these differences may introduce a degree of uncertainty, particularly in comparing composition over time.

Regarding the vegetation data for native plots, the surveys were conducted using a standardized protocol as part of earlier projects (ISLANDBIODIV and MOVECLIM), and are based on peer-reviewed methodology (as cited in Borges et al. 2018, Global Island Monitoring Scheme). Although these data are from 2012–2013, the sites are in well-preserved, minimally disturbed forest areas within protected areas. These forests are known for their structural stability, and currently no more recent peer-reviewed vegetation surveys exist for these exact plots.

Moreover, the wood species composition and abundance from those plots is already available in the supplementary material of the publication below, that we now cite in the methods section.

Borges, P.A.V., Cardoso, P., Fattorini, S., Rigal, F., Matthews, T.J., Di Biase, L., Amorim, I.R., Florencio, M., Borda-de-Água, L., Rego, C., Pereira, F., Nunes, R., Carvalho, R., Ferreira, M.T., Lopez, H., Pérez Delgado, A.J., Otto, R., Fernández Lugo, S., Nascimento, L. de, Caujapé-Castells, J., Casquet, J., Danflous, S., Fournel, J., Sadeyen, A.-M., Elias, R.B., Fernández-Palacios, J.M., Oromí, P., Thébaud, C., Strasberg, D. & Emerson, B.C. (2018). Community structure of woody plants on islands along a bioclimatic gradient. Frontiers of Biogeography, 10(3-4): 1-31. DOI: 10.21425/F5FBG40295

As mentioned, new field sampling in these sensitive areas was avoided to prevent unnecessary disturbance to the forest structure, which is central to our study’s focus on structural complexity. We recognize this is a limitation and have now explicitly acknowledged it in the manuscript.

While we understand the reviewer’s concern, we believe that the combination of (i) the known stability of these forest systems, (ii) the location of plots in protected and undisturbed areas, and (iii) the use of a standardized and peer-reviewed vegetation survey protocol, provides a sufficient basis for using these data, while transparently communicating the limitation to readers.

Material and Methods. Vegetation Survey

Original comment: The exotic forest plots were surveyed in the spring of 2023 within 20 x 20 (…). Why are the plots different in size?

Reply by the authors: The larger 50 x 50 meter plots for the native forests were chosen during the design of the ISLANDBIODIV project and we use the data gathered. The 20 x 20 meter plots in the exotic forests were selected based on forest height, accessibility and logistical considerations. Furthermore, the homogeneity of the plots supported the selection of this area to represent the forest structure being studied.

Homogeneity is a type of data, not shown in the paper, just assumed in the reply, if the authors can show or produce evidence of this

---

## [Editor Report · Decision Letter 2]

28 May 2025

Landscape to microhabitat: uncovering the multiscale complexity of native and exotic forests on Terceira Island (Azores, Portugal)

PONE-D-24-59937R2

Dear Dr. Lhoumeau,

We’re pleased to inform you that your manuscript has been judged scientifically suitable for publication and will be formally accepted for publication once it meets all outstanding technical requirements.

Kind regards,

Francesco Boscutti

Academic Editor

PLOS ONE
---

## [Editor Report · Acceptance letter]

PONE-D-24-59937R2

PLOS ONE

Dear Dr. Lhoumeau,

I'm pleased to inform you that your manuscript has been deemed suitable for publication in PLOS ONE. Congratulations! Your manuscript is now being handed over to our production team.

Kind regards,

on behalf of

Dr. Francesco Boscutti

Academic Editor

PLOS ONE